



# Glaciers and Climate of the Upper Susitna Basin, Alaska

Andrew Bliss[1,2], Regine Hock[2], Gabriel Wolken[3], Erin Whorton[3,4], Caroline Aubry-Wake[2,5],
Juliana Braun[2,6], Alessio Gusmeroli[7], Will Harrison[2], Andrew Hoffman[2], Anna Liljedahl[8], and
Jing Zhang[9]

[1]Department of Anthropology, Colorado State University, 1787 Campus Delivery, Fort Collins, CO 80523, USA
[2]Geophysical Institute, University of Alaska Fairbanks, Fairbanks, AK, USA
[3]Alaska Division of Geological and Geophysical Surveys, Fairbanks, AK, USA
[4]USGS Washington Water Science Center, Tacoma, WA, USA
[5]Centre for Hydrology, University of Saskatchewan, Saskatoon, SK, Canada
[6]Willis Re, London, UK
[7]International Arctic Research Center, University of Alaska Fairbanks, Fairbanks, AK, USA
[8]Water & Environmental Research Center and International Arctic Research Center, University of Alaska Fairbanks,
Fairbanks, AK, USA
[9]Department of Physics and Department of Energy & Environmental Systems, North Carolina A&T State University,
Greensboro, NC, USA

**Correspondence:** Andrew Bliss (andybliss@gmail.com)

**Abstract.** As part of a proposed hydropower facility, extensive field observations were conducted in the Upper Susitna basin, a 13,289 km$^2$ glacierized catchment in central Alaska in 2012-2014. This paper describes a comprehensive data set of meteorological, glacier mass balance, snow cover and soil measurements, as well as the data collection and processing. Results are compared to similar observations from the 1980s. Environmental lapse rates measured with weather stations between about 1000 and 2000 m a.s.l. were significantly lower over the glaciers than the non-glaciated areas. Glacier-wide mass balances shifted from close to balanced in the 1980s to less than -1.5 m w.e. yr$^{-1}$ in 2012-2014. Winter snow accumulation measured with ablation stakes on the glaciers closely matched observations from helicopter-borne radar. Soil temperature measurements across the basin showed that there was no permafrost in the upper 1 m of the soil column. The data produced by this study is available at http://doi.org/10.14509/30138 and will be useful for hydrological and glaciological studies including modeling efforts.

## 1 Introduction

Climate change is projected to have significant impacts on future water resources. In snow- and glacier dominated catchments the response is strongly affected by changes in snow and glacier storage (Bliss et al., 2014; Huss and Hock, 2018). Changes in precipitation amounts and seasonality, air temperature, glacier mass balance, and vegetation type all contribute to changes in river runoff. Understanding present-day relationships among these contributing factors can help improve projections of future river runoff. To address these questions intensive glaciologic and hydrologic fieldwork was conducted in the headwaters of the Susitna River's watershed, central Alaska, during the 1980's in connection with a proposed hydro-electric dam construction on the Susitna River (Clarke, 1991; R&M Consultants and Harrison, 1981; Clarke et al., 1985). The dam was not built,



but when the proposal resurfaced approximately 30 years later (Susitna-Watana Hydroelectric Project, http://www.susitna-watanahydro.org/), we performed extensive field measurements in the same area. Our work combined field measurements with glacier runoff modeling to make projections of the effect of climate change induced future glacier mass changes on the inflow to the proposed dam; this paper focuses on the measurements.

More than 120 glaciers flow down the southern flanks of the central Alaska Range into the three forks of the Upper Susitna River (Figure 1). The glaciers provide a significant portion of the total runoff within the Upper Susitna drainage. It is well documented that glaciers across Alaska are currently retreating (Gardner et al., 2013; Luthcke et al., 2013). Changes to the timing and amount of runoff due to continued melting of glaciers have been projected to occur worldwide (Bliss et al., 2014; Huss and Hock, 2018). Therefore, it is important to understand how changes to the Upper Susitna basin glaciers and river flow
could affect dam operations and environmental resources.

This paper describes the data collected during the 2012-2014 field campaign detailing the instrumentation, method of deployment, and results for each set of data. Observations included meteorological variables, glacier mass balance, snow depth and density, and soil type and temperature. Where possible we also compare the data with the results from the 1980s field campaign.

## 15   2   Study area

The watershed above the proposed Susitna-Watana dam (62.822523°N, 148.538986°W; henceforth referred to as the Upper Susitna basin) covers an area of 13,289 km$^2$ with elevation spanning from 450 to 4,200 m above sea level (a.s.l., Figure 1). About 4% of the basin is glacierized. The total glacier area is 678.4 km$^2$ according to the Randolph Glacier Inventory version 6.0 (Pfeffer et al., 2014; Kienholz et al., 2015), which is based on satellite imagery from 3 July 2009. Modern glaciers are well
within the limit of the Late Wisconsinan glacial advance (20-25 ka BP), when this part of the Alaska Range hosted the northern extent of the Cordilleran Ice Sheet (Kauman and Manley, 2004).

Almost all of the basin's glacier area is found in the Alaska Range whose highest ridges and peaks form the basin's northern boundary. This area is characterized by high relief (Figure 1). Most glaciers (in total 127) in the study area are located in the Alaska Range, but a few small glaciers exist in the Talkeetna Mountains which form the southwest boundary of the basin.

The glacier monitoring work focused on the five largest glaciers in the Alaska Range: West Fork Glacier (193.4 km$^2$), Susitna Glacier (209.6 km$^2$), East Fork Glacier (39.8 km$^2$), Maclaren Glacier (56.5 km$^2$), and Eureka Glacier (34.0 km$^2$). Apart from a former tributary of the West Fork Glacier (33.0 km$^2$), which is now disconnected, the remaining glaciers are smaller than 7 km$^2$. Ninety-three of the 127 glaciers in the Alaska Range are smaller than 1 km$^2$ and their total area is 32.3 km$^2$. Using a volume-area scaling relationship (Bahr et al., 1997), we estimate a total glacier volume of 137 km$^3$ for the Upper Susitna
basin. We use scaling coefficients for mountain glaciers ($c = 0.2055$ m$^{3-2\gamma}$, $\gamma = 1.375$). If we assume an ice density of 900 kg m$^{-3}$, this represents 123 Gt of ice. Some of the larger glacier termini reach elevations between 800 and 900 m a.s.l.

The nine glaciers in the Talkeetna Mountains draining to the Susitna river have a combined area of 8.9 km$^2$. The largest glacier, located at the head of the Black River, is 7.3 km$^2$. The total glacier volume is less than 0.6 km$^3$ (0.5 Gt).





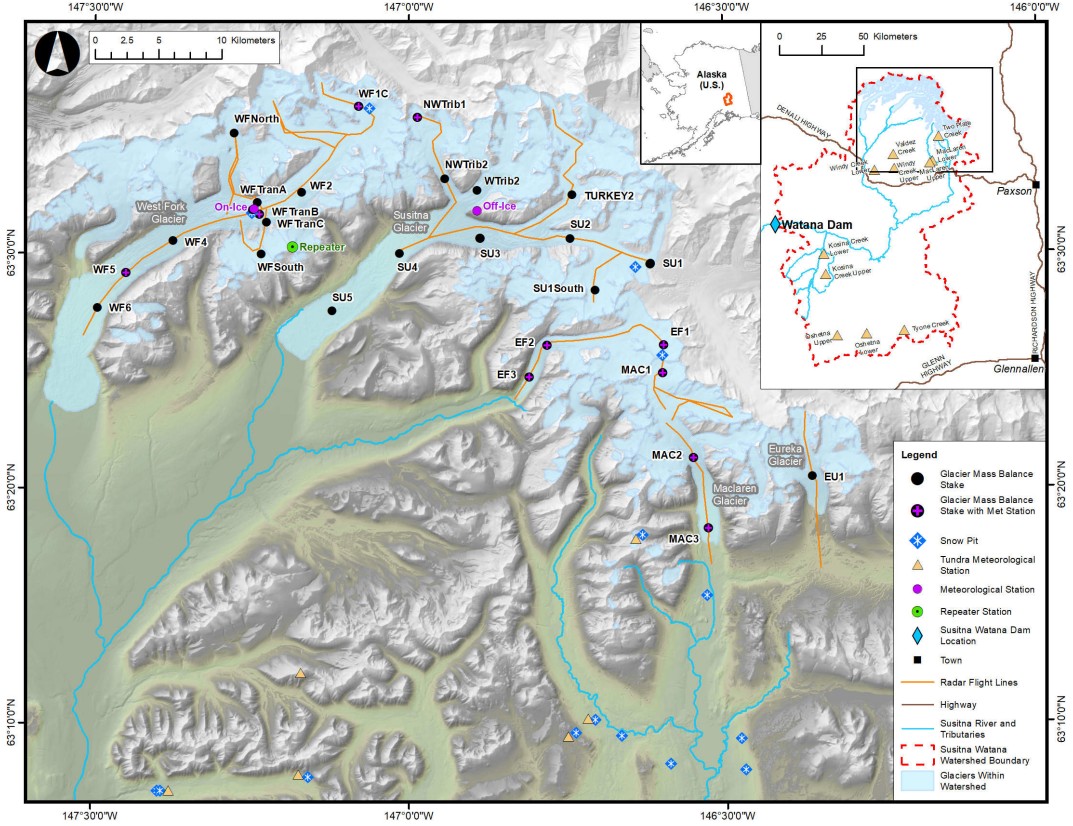

**Figure 1.** Map of study area including measurement locations. From west to east, the major glaciers are West Fork Glacier (stations with the prefix WF), Susitna Glacier (SU), East Fork Glacier (EF), Maclaren Glacier (MAC), and Eureka Glacier (EU).

Significant portions of the large glaciers in the Alaska Range are covered by rock debris. A Landsat image from 15 September 2010 revealed that the disconnected tributary of West Fork Glacier was 7% covered by debris. Debris covered 18% of West Fork Glacier, 26% of Susitna Glacier, 3% of East Fork Glacier, 6% of Maclaren Glacier and 7% of Eureka Glacier. Debris of sufficient thickness, like that found on West Fork and Susitna Glaciers in particular, has an insulating effect on the ice
5    underneath, reducing the amount of ice melt compared to bare ice areas.

Wastlhuber et al. (2017) determined glacier area and mass changes of the basin's glaciers between 1951 and 2010 and found substantial glacier retreat and mass losses. During this period the glaciers lost an area of $128\pm15$ km$^2$ (16%) and thinned on average by $0.41\pm0.07$ m yr$^{-1}$. The average thinning rate almost tripled ($1.20\pm0.25$ m yr$^{-1}$ during the later period 2005-2010. Using satellite imagery, the average equilibrium (1999–2015) line altitude was found to be at $1745\pm88$ m a.s.l..

10    Both Susitna and West Fork glacier have a history of surging. Surge-type glaciers experience episodic acceleration of flow at many times their normal velocities, transferring large amounts of ice to lower elevation, and usually result in rapid terminus advance and outburst floods. The last known Susitna Glacier surge occurred in 1951 or 1952, with a pronounced terminus advance and a maximum ice movement of about 4 km (Post, 1960; Clarke, 1991). West Fork glacier surged in 1935 or 1937,



and again from 1987 to 1988. The latter produced a maximum ice displacement of 4 km and a surface elevation increase of up to 120 m observed near the terminus (Clarke, 1991; Harrison et al., 1994). Harrison et al. (1994) report that during the termination of the 1987-88 surge that runoff and sediment fluxes sharply increased from the glacier to the Susitna river.

During quiescent periods mean annual glacier surface velocities in the Upper Susitna basin are estimated to range from 0 to
0.73 m d$^{-1}$ (Burgess et al., 2013); the highest velocities occur on Susitna and West Fork glaciers. Some glaciers experience brief periods of acceleration in spring, which have been linked to enhanced basal lubrication caused by meltwater (MacGregor et al., 2005; Bartholomaus et al., 2008). Periods of deceleration in late summer have been connected to warm summers and greater meltwater production (Sundal et al., 2011).

The non-glacierized part of the basin is characterized by sparse vegetation and little human development. The southeastern
part of the watershed is characterized by low relief, numerous lakes, and open spruce forest. The largest lakes are Susitna Lake and Lake Louise. A low divide to the south and east separates the Susitna basin from the Copper River and its tributaries.

The majority of the area draining into the proposed dam is estimated to be underlain by discontinuous and continuous permafrost (Figure 2). Maximum depth to the base of permafrost near the Maclaren River junction with the Susitna River is about 200 m (United States Army Corps of Engineers, 1975), while it is 40 m at Gulkana, approximately 50 km southeast of
the basin (Romanovsky et al., 2010).

Nearby weather stations with long-term records include Talkeetna Airport (west of the basin, 1067 m a.s.l.) and Gulkana Airport (east of the basin, 467 m a.s.l.). Annual mean temperature for the period 1985-2014 at Talkeetna was 1.4 ° C and at Gulkana -1.5 ° C (http://ncdc.noaa.gov). Precipitation averaged 710 mm yr$^{-1}$ at Talkeetna and 288 mm yr$^{-1}$ at Gulkana.

Flow of the Susitna River at Gold Creek (62°46'04" N, 149°41'28" W, downstream of the basin considered in this paper)
was 8.8±1.2 km$^3$ yr$^{-1}$ (mean ± standard deviation), or 277.8±36.8 m$^3$ s$^{-1}$, over the period 1950-2015 with a measurement hiatus from 1997-2000. Peak flow was usually in mid-June.

## 3   Climatological and meteorological data

Climate exerts the primary influence on river runoff and glacier mass balance. The meteorological and climatological knowledge of mountainous areas of south-central Alaska, including the Upper Susitna basin, is generally poor, largely due to the
sparse and poorly distributed data (no in-situ data available from high elevations) and the lack of consistent, long-term measurements. To improve the coverage, we strategically placed two energy balance weather stations in the Alaska Range, and 25 simple weather stations throughout the entire watershed (Figures 1 & 3). Table A1 lists the location and elevation of all meteorological stations used in this study.



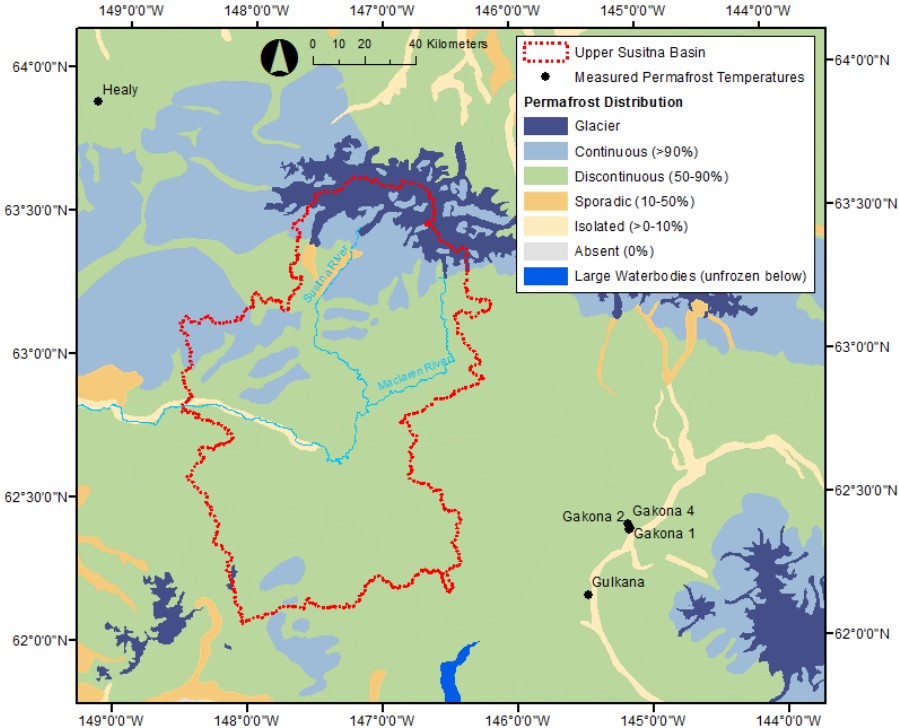

**Figure 2.** Distribution of permafrost in study area (modified after Jorgenson et al. (2008)).

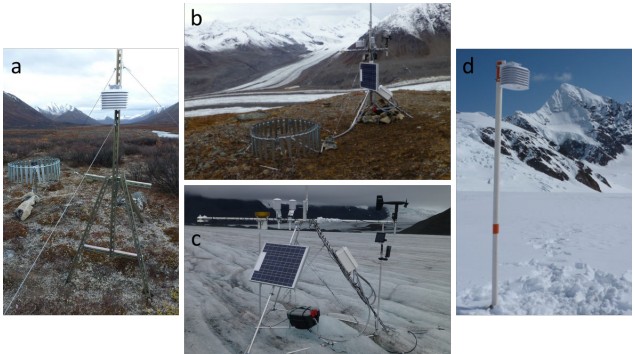

**Figure 3.** Photos of employed weather stations: (a) Example of a tundra station, (b) Off-Ice station perched on a Susitna Glacier nunatak, (c) On-Ice station located on West Fork Glacier, (d) Example of simple glacier station.



## 3.1 Energy balance weather stations

### 3.1.1 Instrumentation

Two energy balance weather stations were installed in the upper part of the basin, one on the West Fork Glacier and one on a ridge between the two branches of the Susitna Glacier (3).

The glacier station (On-Ice) was installed on 16 April 2013 in the glacier's ablation area at 1398 m a.s.l., and operated until 6 September 2014 (Figure 1). The station floated on the ice surface keeping the air temperature sensor at a consistent height as the surface melted (approximately 2 m above the ice surface). All the variables listed in Table 1 were recorded during the summers of 2013 and 2014. Snow temperatures were measured at three levels (0.65 m, 1.15 m, and 1.65 m below the initial snow surface; the snow depth was 2.15 m at installation on 17 April 2013). Sensors were attached to the tip of 20 cm long

aluminum poles which were pushed into the snow (from a snow pit) at a slight upward angle from horizontal to minimize measurement errors caused by the effect of water percolation along cables. Ice temperatures were measured at 14 levels with a thermistor string lowered into a borehole close to the weather station drilled with a hot water drill. The upper seven thermistors were spaced 0.5 m apart, while the remaining ones were 1 m apart. At installation, the uppermost thermistor was at 0.1 m and the lowest one at 10 m below the ice-snow interface. Surface elevation change was measured with a sonic ranger on a separate

pole drilled into the ice 5.6 m north of the weather station. A camera on the weather station mast took pictures of the sonic ranger pole every hour, providing visual information on surface type (snow, ice) and the surface elevation change. The sonic ranger pole was marked with tape at 10 cm intervals for visual clarity.

    For winter 2013/2014, some instruments were removed (radiometers, precipitation gauge, sonic ranger, and camera) and data are not available for the period 4 September 2013 - 25 April 2014 (Figure 4). Based on the relative humidity values at the

On-Ice station and precipitation data from nearby manned weather stations (Alpine Creek Lodge, Talkeetna Airport, Gulkana Airport), we estimate that the station's air temperature and relative humidity sensors were buried by snow from 19 January 2014 until the field visit on 22 April 2014. During the field visit, we excavated the station, placed it on top of the winter snowpack and reinstalled the instruments removed for winter.

    A second multi-variable weather station (Off-Ice) was installed at 1516 m a.s.l., 18 km east of the On-Ice station (Figure

1). The station records all the variables listed in Table 1 except for snow and ice temperatures and snow surface elevation changes. The station was installed on 16 July 2012 and continues to operate under DGGS stewardship (here we present data through 6 September 2014). A low battery caused the station to stop working on 24 March 2013 but it was restarted on 16 April 2013. Liquid precipitation data are not continuous during the 2012-2014 record since the rain gauge was removed each fall and reinstalled in spring.

Both multi-variable stations sampled their sensors every minute and recorded averages (or sums in the case of precipitation) for hourly and daily values. Wind speed and direction were sampled on a 3-second interval to capture maximum wind speeds during gusts, yet average wind speeds were recorded for hourly and daily intervals. Barometric pressure was sampled every 30 minutes instead of every minute.



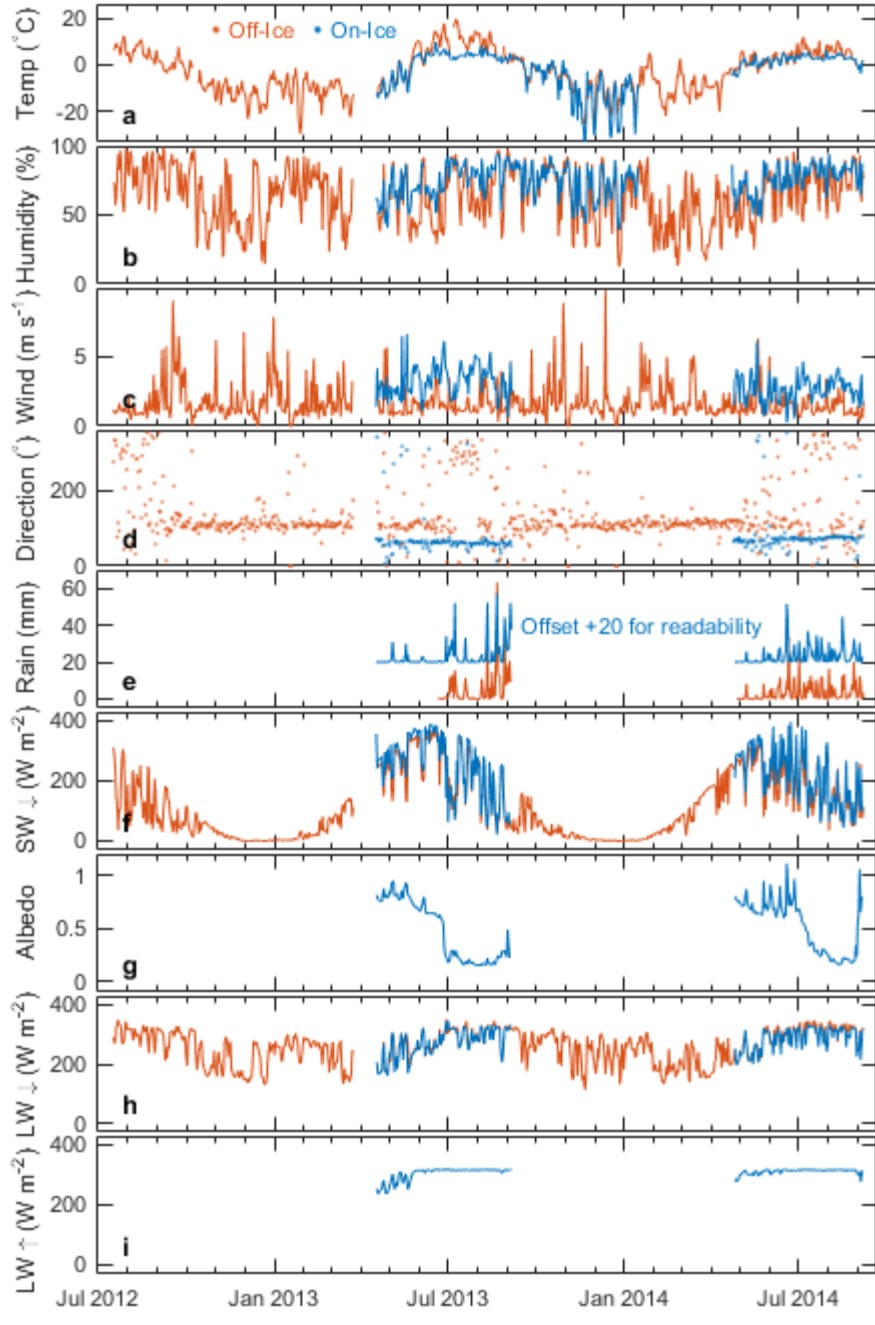

**Figure 4.** Meteorological data from the On-Ice and Off-Ice stations from July 2012 to September 2014. (a) Daily temperature, (b) relative humidity, (c) wind speed, (d) wind direction, (e) precipitation, (f) incoming shortwave (SW) radiation, (g) albedo, (h) incoming longwave (LW) radiation, and (i) outgoing longwave radiation. All values are daily means except for precipitation which is the daily sum. Data gaps occur in winter when some sensors were removed. The On-Ice temperature/humidity sensor was buried in the snowpack in mid-January 2014 and re-exposed during a station visit on 22 April 2014 so data are not included for that interval.



**Table 1.** Sensors used for the energy balance weather stations (On-Ice and Off-Ice).

| Variable | Sensor | Unit | Accuracy |
|---|---|---|---|
| Temperature | Rotronic HygroClip2 Temperature/RH Probe* | $^\circ$C | 0.1$^\circ$C |
| Relative humidity | Rotronic HygroClip2 Temperature/RH Probe* | % | 0.8% |
| Barometric pressure | Vaisala PTB110 Barometer | hPa | 1 hPa |
| Incoming longwave radiation | Hukseflux 4-Component Net Radiation | W m$^{-2}$ | 10% |
| Outgoing longwave radiation | Hukseflux 4-Component Net Radiation | W m$^{-2}$ | 10% |
| Incoming shortwave radiation | Hukseflux 4-Component Net Radiation | W m$^{-2}$ | 10% |
| Outgoing shortwave radiation | Hukseflux 4-Component Net Radiation | W m$^{-2}$ | 10% |
| Rainfall | Texas Electronics Rain Gage** | mm | 1% |
| Tilt of the radiation sensors | Turck Inclinometer B2N45H-Q20L60-2LU3-H1151 | degrees | 0.5 degrees |
| Wind direction | RM Young Wind Monitor, Alpine version | degrees | 5 degrees |
| Wind speed | RM Young Wind Monitor, Alpine version | m s$^{-1}$ | 0.3 m s$^{-1}$ or 1% |
| Distance to ice surface (ablation)*** | SR50A | m | 1 cm or 0.4% |
| Snow temperature*** | Thermistor 3K Ohm from Digikey | $^\circ$C | 0.1 $^\circ$C |
| Ice temperature*** | Thermistor 3K Ohm from Digikey | $^\circ$C | 0.1 $^\circ$C |
| Datalogger | Campbell Scientific CR1000 | - | - |

\* Shielded with a RM Young 10-plate Gill shield.

\*\* Installed at the Off-Ice station at 0.6 m (top of gauge to ground surface) and shielded with a Novalynx Alter-type Rain Gage Wind Screen. Installed at the
On-Ice station on top of the instrument arm at 2 m above the surface and unshielded (Figure 3). Gauges were not heated and hence gave most accurate results when
precipitation was liquid rather than solid.

\*\*\* Only at the On-Ice station

### 3.1.2 Meteorological data

The daily meteorological data for both the On-Ice and Off-Ice station are shown in Figure 4, and seasonally averaged correlation
coefficients between the two station's daily data are given in Table 2. Air temperatures at each station were significantly more
variable in winter than in summer due to frequent winter weather systems. Both station's daily temperatures correlated well
5 in all seasons except summer (June-August), when temperatures at the On-Ice station exhibited considerably less day-to-day
variability than the Off-Ice station. In addition, albeit 118 m lower in elevation, the On-Ice station's temperatures were also
more than 4$^\circ$C lower than at the Off-Ice station. These differences are attributed to the fixed glacier surface temperature of
0$^\circ$C during the extended periods of glacier melting during summer. The cold glacier surface cooled the air above it. Though
differences were not as pronounced, lower air temperatures at the On-Ice station were also observed during other seasons. This
10 is due, in part, to the high albedo at the On-Ice station.

Relative humidity at the On-Ice station was typically higher than at the Off-Ice station, consistent with lower air temperatures
On-Ice and greater availability of moisture for evaporation. The On-Ice station also displayed less day-to-day variability than
at the Off-Ice station. Data are not available from from 18 January 2014 to 22 April 2014 when the station was buried in snow.



**Table 2.** Seasonal means of meteorological variables for the On-Ice and Off-Ice stations based on data between 18 April 2013 and 6 September 2014. All data are for times when both stations were operating, so these data do not represent full season averages. Data were excluded from the period when the On-Ice station was buried by snow: 18 January 2014 to 22 April 2014. Correlations for precipitation excluded days when both stations had no precipitation.

| Variable | | Range | DJF | MAM | JJA | SON |
|---|---|---|---|---|---|---|
| Temperature $(°C)$ | On-Ice | -33.6 to 9.7 | -15.2 | -3.1 | 3.5 | -5.9 |
| | Off-Ice | -25.6 to 19.5 | -9.7 | -4.6 | 7.2 | -5.0 |
| | r | | 0.91 | 0.93 | 0.64 | 0.96 |
| Relative humidity $(\%)$ | On-Ice | 39.0 to 94.4 | 70.0 | 65.6 | 79.2 | 75.2 |
| | Off-Ice | 12.9 to 97.7 | 54.6 | 53.7 | 69.3 | 71.6 |
| | r | | 0.91 | 0.68 | 0.84 | 0.93 |
| Wind speed $(ms^{-1})$ | On-Ice | 0.3 to 6.5 | $\cdots$ | 2.8 | 3.1 | 2.5 |
| | Off-Ice | 0.0 to 9.8 | 1.9 | 1.6 | 1.5 | 1.8 |
| | r | | $\cdots$ | 0.23 | 0.26 | 0.83 |
| Wind speed max $(ms^{-1})$ | On-Ice | 2.1 to 16.8 | $\cdots$ | 7.5 | 6.9 | 8.3 |
| | Off-Ice | 0.0 to 30.6 | 8.1 | 7.3 | 7.2 | 8.4 |
| | r | | $\cdots$ | 0.67 | 0.64 | 0.78 |
| Precipitation $(mm)$ | On-Ice | 0.0 to 37.4 | $\cdots$ | 0.8 | 3.9 | 8.9 |
| | Off-Ice | 0.0 to 63.2 | $\cdots$ | 0.8 | 4.7 | 7.6 |
| | r | | $\cdots$ | 0.86 | 0.72 | 0.63 |
| Incoming shortwave $(Wm^{-2})$ | On-Ice | 21 to 397 | $\cdots$ | 286 | 217 | 129 |
| | Off-Ice | 0 to 369 | 16 | 215 | 200 | 49 |
| | r | | $\cdots$ | 0.93 | 0.97 | 0.99 |
| Incoming longwave $(Wm^{-2})$ | On-Ice | 161 to 345 | $\cdots$ | 237 | 295 | 293 |
| | Off-Ice | 113 to 351 | 225 | 227 | 301 | 265 |
| | r | | $\cdots$ | 0.94 | 0.94 | 0.94 |

Daily mean wind speeds were typically higher at the On-Ice station than at the Off-Ice station which can be attributed to the relatively smooth glacier surface, longer fetch, and summer-time katabatic wind. Highest wind speeds occurred during winter with maximum wind gusts (3-second averages) up to 30.5 m s$^{-1}$ (11 December 2013).

Total precipitation at the Off-Ice station was slightly higher (9%) than at the On-Ice station during the periods both stations were functional, however, direct comparison is difficult since the instrumental set-up was different with no wind shield and installation higher above the surface at the On-Ice station.

Incoming solar radiation displayed pronounced seasonal variability consistent with the site's latitude close to the polar circle. Daily mean values varied between just 16 W m$^{-2}$ in winter and approximately 400 W m$^{-2}$ in summer. Cloudy days in summer



are clearly discernible due to their lower shortwave radiation compared to neighboring sunny days. High correlation (r=0.97) between both daily time series indicates relatively homogeneous cloud conditions at those sites. A portion of the difference between the two stations may be due to topographic shading.

In both 2013 and 2014, daily albedo in May and June had peaks between 0.7 and 1 and by late July and August albedo fell to

0.2. The summer of 2013 featured a drop in albedo from 0.6 to 0.2 over 5 days starting 25 June 2013 indicating the transition of a snow-covered to a snow-free surface. In 2014, the same transition took one month starting 4 July 2014. We did not measure reflected shortwave radiation at the Off-Ice station, but expect it to be significantly lower than the On-Ice station due to the low albedo of the rocky surface compared to the On-Ice glacier surface.

Incoming longwave radiation, which depends on the effective radiative temperature of the atmosphere, showed a slight

seasonal cycle (Off-Ice station) with an amplitude of roughly 80 W m$^{-2}$. Large daily variations superimposed on the seasonal cycle and negatively correlated with incoming solar radiation indicate variations in cloud cover. Incoming longwave was well-correlated between the On-Ice and Off-Ice stations (r=0.94 for summer).

In both summers (2013 and 2014), outgoing longwave radiation increased through the spring to just below 320 W m$^{-2}$ in mid-June and then plateaued through the end of August. Blackbody radiation from an object at 0 °C would be expected to be

316 W m$^{-2}$, indicating that the effective radiative temperature of the ice surface and air between the surface and the sensor was just above 0 °C, and the surface was melting uninterruptedly for extended periods in summer.

### 3.1.3 Snow and ice temperatures

The thermistor string we deployed in a hot-water-drilled borehole needed time to equilibrate to its surroundings after installation. We consider the thermistor ice temperature measurements to be reliable starting on 25 April 2013, about 8 days after

installation. By that time, the 5 m deep thermistor temperatures were within 0.02 °C of the trend they held for the subsequent 4 weeks.

Temperatures within the upper 10 meters of the glacier surface ranged from approximately -10°C in the upper layers of snow pack in early May 2013 to close to the melting point of 0°C at 10 m below the ice surface (Figure 5).

As air temperatures rose, the subsurface temperatures of the upper layers increased but with a time lag that increased with

depth (Figures 5 and 6). When the air temperature rose above 0°C, surface melt began to occur. As meltwater or rain percolated into the snowpack it refroze, causing abrupt temperature increases of the uppermost thermistors (e.g. 10 and 25 May 2013, Figure 5). The glacier surface lowered with respect to the subsurface sensors by roughly 5 m between late April and early September 2013 as the melt season progressed (Figure 5b). The three snow sensors and uppermost 6 ice sensors were exposed to the air sequentially during the summer melt season. On 24-25 June, the ice temperature sensors at 0.1, 0.5, 1, and 1.5 m

below the snow-ice interface all experienced rapid warming to about 0°C. The 1.5 m sensor then cooled back down to -0.85°C. We interpret this to be another meltwater event but this time in the ice rather than the snowpack. It is difficult to know whether the event was representative of conditions in the glacier (i.e. water moving through cracks and along grain boundaries in the ice) or simply conditions along the thermistor cable. By 27 June, all 2.1 m of snow at the site had melted and the ice surface was exposed, as measured by the SR50 and confirmed with the time lapse imagery. After 27 June, sensors at 2, 2.5, and 3





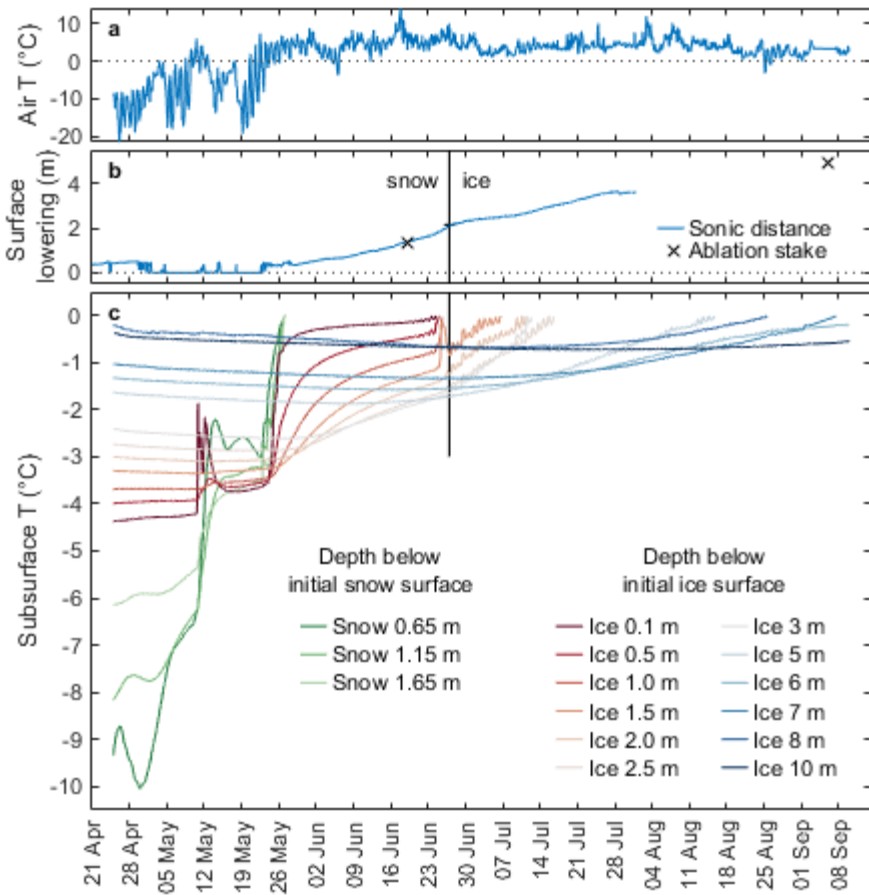

**Figure 5.** Hourly ice and snow temperature (T) measurements from thermistors installed at depth in the snow and ice adjacent to the On-Ice weather station (c). Panel (a) shows air temperature, (b) shows cumulative surface lowering measured by the sonic ranger and ablation stakes. Initial thermistor depths are listed in the legend given in depth below the initial snow surface for the three snow thermistors and depth below the snow-ice surface interface for the 12 ice thermistors. The snow depth at installation on 19 April 2013 was 2.10 m. Note that the instrumentation depth becomes shallower as the glacier surface ablates during the melt season. The onset of large diurnal temperature fluctuations above 0°C indicates that the thermistors have melted out and are affected by solar radiation. After a sensor exceeds 0°C, we do not plot the remaining data.

m depth exhibited diurnal temperature wiggles with an amplitude of up to 0.2°C. By mid-July the diurnal wiggles appeared in the sensors at 5, 6, and 7 m depth too. Some of the ice temperature sensors at depths ⩾5 m recorded temperatures greater than 0°C starting in late August 2013. These measurements are likely errors, perhaps due to faulty voltage sensor outputs. The data-logger continued to record reasonable results for other variables.





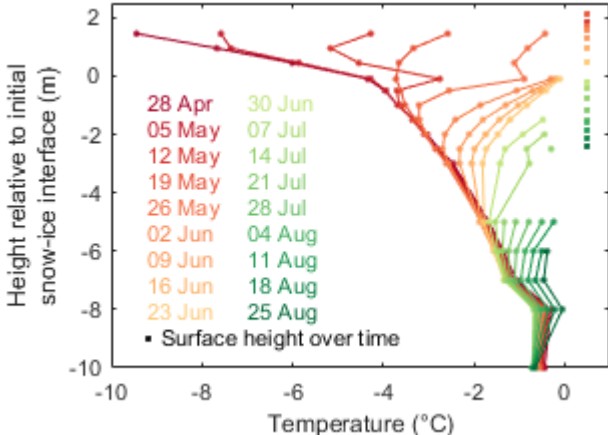

**Figure 6.** Vertical profiles of temperature in the upper 10 m of the glacier at the On-Ice station in summer of 2013. After a sensor exceeds $0°C$, we do not plot the remaining data.

## 3.2 Simple weather stations

### 3.2.1 Instrumentation

To supplement the multi-variable weather stations described above and constrain the spatial patterns of temperature and precipitation within the basin, we installed 26 simple weather stations across the basin both on and off the glaciers (Figure 1; Table A1). The 14 stations on or very near the glaciers (EF1, EF2, EF3, Mac1, Mac2, Mac3, Repeater HOBO, NWTrib1, Off-Ice HOBO, SU1, SU3, WF1, WFTranB, WF5; letters refer to glacier names) measured air temperature and relative humidity at a nominal height of 1.75 m above the glacier surface. We refer to these stations collectively as the "glacier" weather stations. The sensor mounts for the glacier stations were designed to maintain approximately the same sensor height relative to the glacier surface throughout the ablation season. This was accomplished by allowing the mount to slide down the ablation stake as the glacier surface melted (Figure 3). The other 12 simple weather stations are referred to as the "tundra" stations. The typical tundra station measured temperature, relative humidity, rainfall, and soil temperature at 10 cm and 1 m depths. Simple weather station instruments are listed in Table 3.

Temperature and relative humidity were recorded every 10, 15, 30 minutes (most stations), or 60 minutes, depending on the station. Each station's data was then averaged in post-processing to hourly and daily values. Hourly and daily precipitation sums were calculated for each tip of the rain gauge tipping bucket.

### 3.2.2 Sensor calibration

Eight factory-calibrated HOBO temperature and relative humidity sensors were co-located with a factory-calibrated Rotronic sensor connected to a Campbell datalogger in April 2013 in Fairbanks, Alaska, for calibration. To avoid errors due to solar



**Table 3.** Sensors used for the simple glacier and tundra weather stations. Glacier stations included only temperature and relative humidity while tundra stations also included precipitation and soil temperature measurements.

| Variable | Sensor | Unit | Accuracy |
|---|---|---|---|
| Temperature | HOBO Pro v2 U23-001 * | °C | 0.21 °C |
| Relative humidity | HOBO Pro v2 U23-001 * | % | 3.5% |
| Rainfall | HOBO RG3-M ** | mm | 1% |
| Soil temperature | HOBO Pro v2 U23-003 2x External Temp. | °C | 0.21 °C |

\* shielded with a HOBO M-RSA Gill-type shield.

\*\* shielded with a Novalynx Alter-type Rain Gage Wind Screen.

radiation the sensors were installed in an outside area shaded during most hours, in particular during mid-day. Another set of 14 HOBO sensors were calibrated at the same location in April 2014. Measurements were done every 5 minutes and hourly averages were logged. Temperatures during the 2013 calibration period ranged from -22 °C to -5 °C and from -15 to +10 °C in 2014 (Figure 7). The temperature offset of hourly-mean HOBO sensors relative to an arbitrarily-chosen reference HOBO

station was typically within ±0.1 °C and rarely beyond ±0.3 °C. Over both periods, the temperature offset had a mean of 0.02 °C and a standard deviation of 0.07 °C (Table 4).

Compared to the Campbell/Rotronic sensor, HOBO temperatures were lower by 0.2-0.3 °C on average °C (Table 4) but differences exhibited a diurnal cycle with temperatures more than 1 °C lower during mid-day in many cases (Figure 4). Five-minute data showed that the response time of the HOBO sensors was slower than the Campbell sensors, but this can not explain

the differences in hourly or daily means (Figure 8). The correlation coefficient between hourly values of temperature from the Rotronic sensor and HOBO sensors was 1.0.

The lack of a consistent pattern in these comparisons prevented us from adjusting the HOBO temperature data to match the Campbell data. The high correlation and low temperature offsets among sensors gave us confidence that using HOBO stations to assess temperature patterns across the basin was valid.

Measured relative humidity ranged from approximately 25% to 90% during the April 2013 calibration period and about 20% to 90% in April 2014 (Figure 7). HOBO sensors generally recorded higher relative humidity compared to the Rotronic sensor (Table 4). The offset in temperature (HOBO was colder than Campbell) explains part of that difference, though absolute humidity calculations show that the HOBO sensors are registering higher total moisture content for both the cold (2013) calibration period and the warmer one (2014). Relative humidity values from the HOBO sensors were well-correlated (r=0.99)

with those from the Rotronic (Figure 8).

In mid-April 2014, a laboratory calibration of the tundra station precipitation tipping bucket gauges revealed an intermittent instrument signal problem. Single tips in the rain gauge were being recorded as two tips on the datalogger; thus the recorded precipitation amount appeared twice as much as the actual rainfall amount. In the field data, all HOBO stations recorded some of these double tips. A few tundra stations recorded up to 10% double tips (Kosina Creek Upper, Kosina Creek Lower, and

25 Windy Lower). Precipitation rates that could cause two tips within 2 seconds (360 mm/hr) far exceed the actual precipitation



**Table 4.** Statistics of air temperature and humidity sensor calibration. Eight HOBO sensors were calibrated in 2013 (5-8 April) and 14 sensors in 2014 (15-18 April). For each year the differences in hourly means between each HOBO station and a reference station were calculated; then the differences from all stations were concatenated before calculating the mean, standard deviation, range, and skewness of the distribution. Two reference stations were used: a Rotronic HygroClip2 Temperature/RH Probe measured by a Campbell datalogger and an arbitrarily chosen HOBO station (data in parenthesis).

| Variable | Year | Mean | $\sigma$ | Range | Skewness |
|---|---|---|---|---|---|
| Temperature (°C) | 2013 | -0.3 (0.0) | 0.2 (0.1) | 1.1 (0.6) | -0.6 (-1.1) |
| Temperature (°C) | 2014 | -0.2 (0.0) | 0.3 (0.2) | 2.2 (1.8) | -1.1 (3.0) |
| Relative humidity (%) | 2013 | 5.6 (1.4) | 3.4 (1.2) | 19.4 (10.5) | 1.2 (2.2) |
| Relative humidity (%) | 2014 | 2.2 (-1.0) | 2.0 (1.0) | 13.2 (6.2) | 0.9 (-0.7) |
| Vapor pressure (hPa) | 2013 | 9.3 (3.3) | 6.0 (2.9) | 50.0 (27.3) | 2.4 (2.1) |
| Vapor pressure (hPa) | 2014 | 14.5 (-8.5) | 11.1 (8.2) | 65.3 (54.1) | 0.1 (-0.7) |

rates observed in the basin. To correct this data problem, for every pair of tips that occurred within 2 seconds, we zeroed out the second tip. The analysis done in this paper uses the corrected rainfall data.

We looked for similar double tips in the On-Ice and Off-Ice station data, but did not find any. The HOBO and Campbell sensors use the same internal electronics to detect tips, so we surmise that the Campbell logger is filtering out any double tips
before recording the data.

### 3.2.3   Spatio-temporal variability of air temperature, humidity, and precipitation

Time series of daily air temperature, humidity and precipitation of all stations (On-Ice, Off-Ice and tundra stations) are shown in Figure 9.

Cumulative measured precipitation amounts for the summers 2012, 2013, and 2014 varied significantly across the domain,
while the timing of events was generally consistent among the stations (Figure 9). Precipitation amounts did not correlate significantly with elevation, slope, aspect, or location.

Air temperatures at all sites showed similar seasonal variations but as expected temperatures varied with elevation. Figure 10 shows largely linear trends of monthly mean temperature with cold temperatures at high elevations in summer of 2013 and 2014. Environmental temperature lapse rates showed a larger temperature gradient for the tundra stations compared to the
glacier stations. In summer, when air temperatures were above freezing, the ice surface cooled the air over the glacier. This cold air descended due to its greater density than the air around it, setting up a katabatic flow. Continued cooling from the glacier partially offset adiabatic warming as the air descended and led to the lower temperature gradient at the glacier stations.

Given the difficulty of calculating monthly lapse rates due to missing data (Figure 10), we also calculated weekly average temperature and plotted it to illustrate the changing lapse rate with the seasons across the basin (Figure 11). The summer
glacier/tundra pattern transitioned back to a single lapse rate when the air temperature fell below 0°C. Winter inversions (with




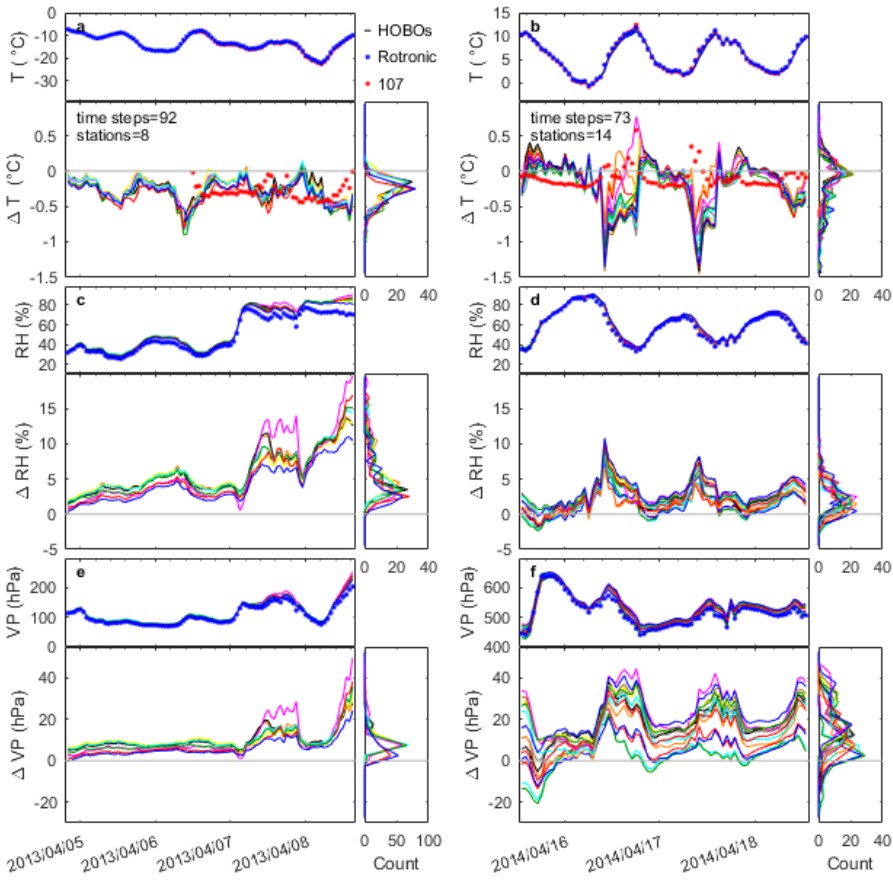

**Figure 7.** Time series of hourly air temperature *T* (a, b), relative humidity *RH* (c, d), and derived vapor pressure *VP* (e, f) of eight HOBO sensors during the calibration period in 2013 (a, c, e) and 14 sensors in 2014 (b, d, f), and their differences to the Rotronic reference sensor. Also shown is the count of differences between HOBO and reference values in bins of 0.1 °C, 1 % and 5 hPa.

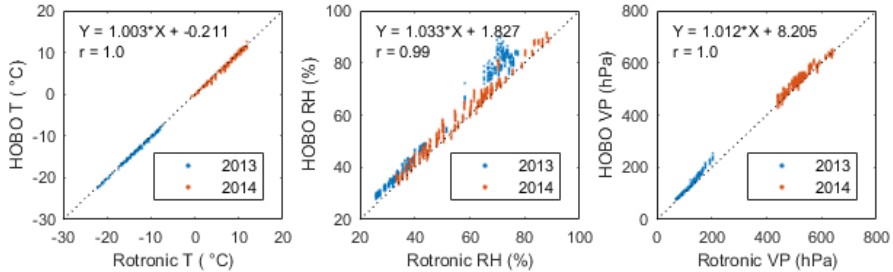

**Figure 8.** Hourly-averaged data from all HOBO sensors compared to the Rotronic sensor for the calibration periods in 2013 (5-8 April) and 2014 (15-18 April). Equations of the best fit lines and correlation coefficients are listed in each panel.



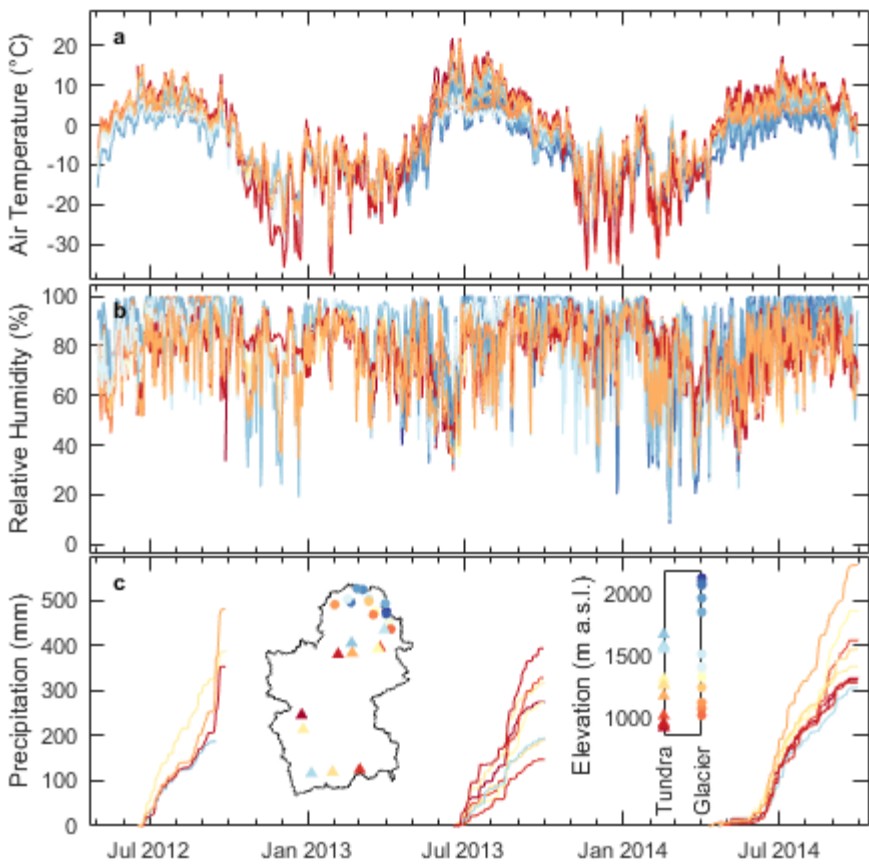

**Figure 9.** Daily mean temperature (a), relative humidity (b), and cumulative precipitation (c) for the tundra and glacier stations. Data is colored by station elevation (inset graph). The inset map shows station locations.

warmer air at higher elevation stations) are common, such as in January 2013 and January 2014. The glacier station at 1400 m a.s.l. was buried by snow for part of each winter and therefore stands out as a warm outlier during those times. Springtime lapse rates are the least complicated - most stations fall along the best fit gradient.

### 3.3    Trends

5    Our field measurements did not cover enough time to establish long term trends of temperature or precipitation. For context, we evaluated nearby stations with long term records from the National Climatic Data Center. The best correlations with our On-Ice station, for temperature and precipitation, came from Talkeetna Airport. The Talkeetna station was 197 km southwest of the On-Ice station and almost 1300 m lower, at 107 m a.s.l. Summer temperatures at Talkeetna Airport in 2012-2014 were 1.1°C warmer than 1981-1983. Annual temperatures were 0.5°C warmer in the recent period. Precipitation was variable year



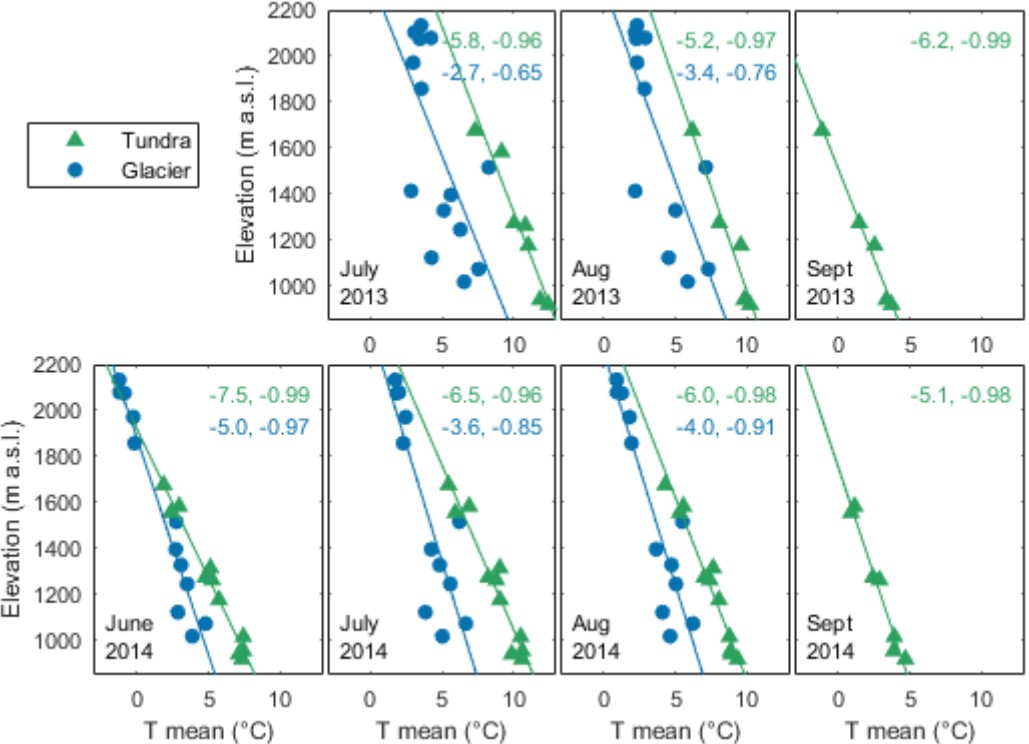

**Figure 10.** Monthly temperature profiles from simple weather stations on the glaciers and on the tundra. Environmental lapse rates (°C km$^{-1}$) were calculated by linear regression between monthly average data and station elevations and are reported in the upper right of each subplot, along with the r-value of the fit.

to year, but showed little trend. Annual precipitation for the three-year interval in the 1980s averaged 746 mm and in the 2010s it averaged 747 mm.

## 4 Glaciological data

Glacier mass changes were determined from in-situ point observations in spring and fall of each year, snow radar measurements

5   in spring and continuous measurements of relative surface elevation change at the On-Ice Weather station.

### 4.1 In-situ point mass balance measurements and derived glacier-wide balances

Winter, summer and annual mass balances were measured at 27 to 29 locations spread across the five largest glaciers, West Fork, Susitna, East Fork, Maclaren, and Eureka (Figure 1) using the glaciological method (Cuffey and Paterson, 2010). The stake distribution was designed to reoccupy the approximate stake positions used in the Clarke et al. (1985) study covering

10   1981-1983, and to sample the elevation range of each glacier. Most measurement sites followed the centerline of glaciers. We



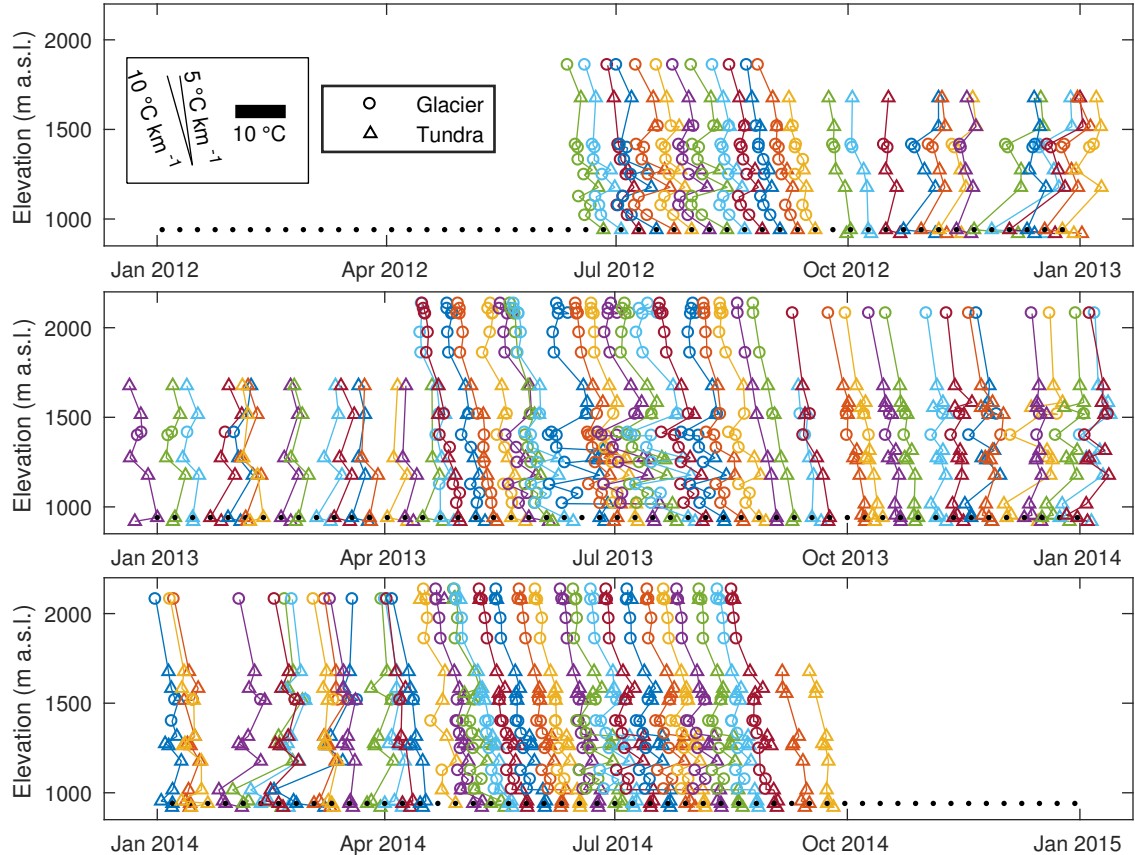

**Figure 11.** Weekly air temperature profiles show the winter inversions and summer differences between glacier and tundra temperature. Each week's profile is plotted relative to a reference station (Windy Creek Lower, 940 m a.s.l.).

performed measurements for three mass balance years: 2012, 2013, and 2014, where a mass balance year refers to the period from October to the following September. For example, year 2012 refers to October 2011 to September 2012. Winter snow accumulation on glaciers was estimated at each stake by snow probing and/or snow pit measurements in late April of each year. The amount of snow and ice that melted each summer was measured at each stake in early September.

5   The most negative annual point balance among the measured glaciers was almost -6 m w.e. (2013 at 1100 m a.s.l. on East Fork Glacier), while the most positive was nearly 2 m w.e. (2014 at 2000 m a.s.l. on Susitna Glacier). Figure 12 indicates a strong elevation-dependence of all point balances. The equilibrium line altitude for the region varied was about 1730 m a.s.l. in 2012 and 1960 m a.s.l. in 2013, as estimated from the zero-crossing of the best-fit line to the mass balance profile data (Figure 12). With a large fraction of the glacier at higher elevations (Figure 12) this corresponds to an accumulation area ratio of 0.58

10  for 2012 and 0.34 for 2013. Wastlhuber et al. (2017) used Landsat imagery to derive equilibrium line altitudes for the set of 5 large glaciers of about 1675 m a.s.l. for 2012 and 1800 m a.s.l. for 2013. Their 2012 value matches ours within uncertainties,



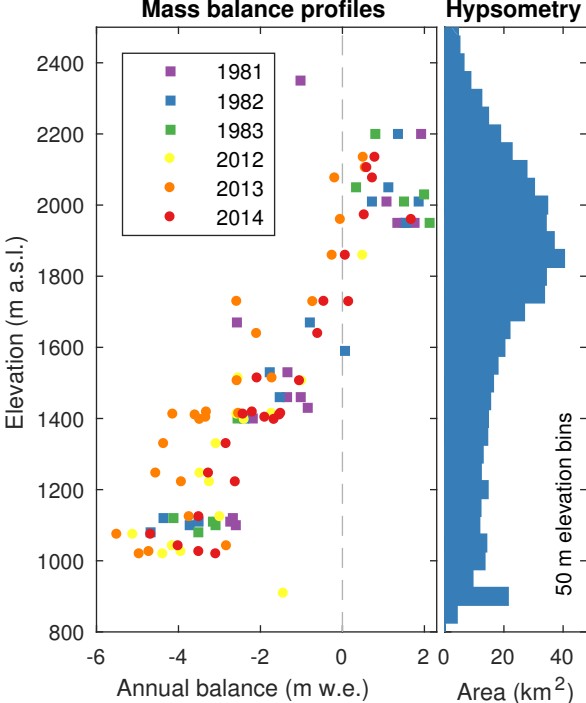

**Figure 12.** Annual point mass balances on West Fork, Susitna, East Fork, Maclaren, and Eureka Glacier versus elevation. The apparent outlier in 2012 (-1.4 m w.e. at 908 m a.s.l.) was due to debris cover at the site. The hypsometry of all 127 glaciers in the Alaska Range portion of the Susitna basin is also shown in 50 m elevation bands. The hypsometry comes from the Randolph Glacier Inventory (Pfeffer et al., 2014), which used glacier outlines from 2009 and elevation from airborne interferometry (http://ifsar.gina.alaska.edu) collected in summer 2010.

but our 2013 value is significantly higher, indicating the surface mass balance might not have been accurately estimated from satellite data.

The point mass balance measurements were used to compute the glacier-wide winter, summer and annual mass balance for each of the five glaciers for the mass balance years 2012, 2013, and 2014. First, we linearly interpolated the mass balance mea-
5    surements across elevations to get a continuous mass balance profile for each glacier. Glacier elevations above and below the measurement range were assigned the mass balance value of their nearest neighbor. The mass balance profiles were distributed to the entire glacier area based on the glacier hypsometry (50 m elevation bins). Glacier-wide mass balance estimates were then calculated by summing the distributed mass balance over the whole glacier. Annual balance was calculated from the sum of winter and summer balance (Table 5). Winter glacier-wide balances ranged from 0.74 m w.e. to 1.3 m w.e.. Glacier-wide
10   summer balances ranged from -4.42 to -1.81 m w.e. The summer balance in 2014 was less negative than in 2012 and 2013. Annual balances ranged from -0.71 to -3.67 m w.e. The glacier-wide annual balance was negative for all years and all glaciers,





**Table 5.** Glacier-wide winter, summer, and annual mass balance (m w.e.) on the five largest glaciers in the Upper Susitna basin, 2012-2014. Mass balances were measured during site visits in the following date ranges: 26 April to 2 May 2012, 26-28 September, 15 -21 April 2013, 6-15 September 2013, 22-27 April 2014, and 7-9 September 2014. Exact dates for individual stake measurements are listed in the dataset.

| Glacier Name | Winter | | | Summer | | | Annual | | |
|---|---|---|---|---|---|---|---|---|---|
| | 2012 | 2013 | 2014 | 2012 | 2013 | 2014 | 2012 | 2013 | 2014 |
| West Fork Glacier | 0.86 | 0.85 | 0.97 | -2.96 | -3.35 | -2.28 | -2.10 | -2.50 | -1.31 |
| Susitna Glacier | 0.88 | 0.60 | 1.17 | -1.95 | -2.37 | -2.10 | -1.07 | -1.77 | -0.93 |
| East Fork Glacier | 0.74 | 1.04 | 1.30 | -4.17 | -2.55 | -2.04 | -3.43 | -1.51 | -0.74 |
| Maclaren Glacier | 0.94 | 1.17 | 1.09 | -3.86 | -2.88 | -1.81 | -2.92 | -1.70 | -0.71 |
| Eureka Glacier | $\cdots$ | 0.74 | 0.89 | $\cdots$ | -4.42 | -3.32 | $\cdots$ | -3.67 | -2.43 |

indicating that the summer mass loss was greater than the winter accumulation. The annual balance in 2014 was less negative than in 2012 and 2013. East Fork Glacier had a similar mass balance as Maclaren Glacier.

### 4.2 Glacier mass balance: comparison to 1980's data

The glacier mass balances changed substantially from 1981-1983 to 2012-2014 (Figure 12). Our measurements for 2012-2014
show that the glaciers in the early 1980's generally had more positive or less negative mass balance than the early 2010's across all elevations (Figure 12). Year 2013 had the most negative mass balance of the six years measured and the highest equilibrium line.

     Consequently glacier-wide summer mass balances became more negative between the 1980s (-0.81±0.21 m water equivalent, mean±std of 4 glaciers) and 2010s (-2.69±0.77 m w.e. mean±std of the same 4 glaciers with stakes in approximately the
same locations). In contrast, winter balance was similar between the 1980s (0.88±0.22 m w.e.) and 2010s (0.97±0.20 m w.e.). Therefore, lower annual balance in the latter period (-1.72±0.87 m w.e.) compared to the former period (0.04±0.25 m w.e.) were driven by the more negative summer balances.

### 4.3 Radar-derived accumulation in glacierized terrain

Unlike ablation, which tends to be spatially coherent and well correlated with elevation, snow accumulation typically shows
pronounced small-scale variability, making it difficult to accurately measure and model (Sold et al., 2013). This is especially true in complex terrain, where topography and meteorological processes vary over short distances (McGrath et al., 2015).

     To robustly validate model simulations of snow accumulation, we conducted helicopter-borne ground penetrating radar (GPR) common-offset surveys of snow accumulation over the five main glaciers and glacier foreland areas in the Upper Susitna basin following Gusmeroli et al. (2014) (Figure 13), and using in-situ measurements of snow density to calculate end of winter
snow water equivalent (SWE) for each year during the period 2012-2014. Radar-derived estimates of winter snow accumulation



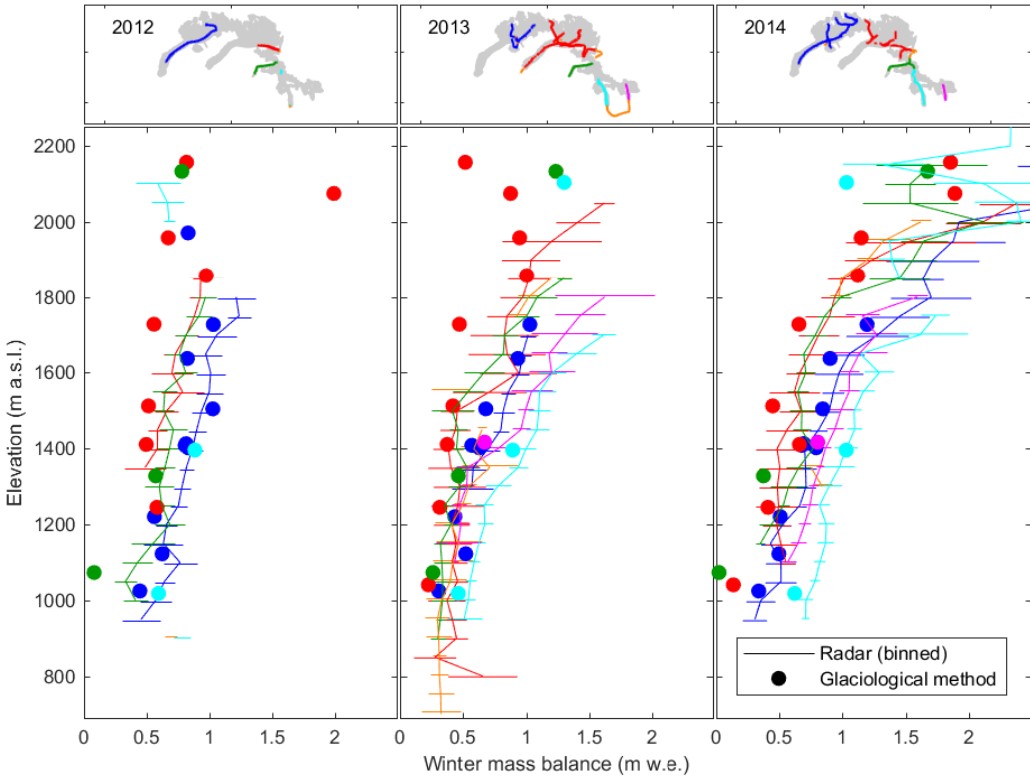

**Figure 13.** Winter mass balance profiles in 2012-2014 derived from point mass balance measurements (filled symbols) and from helicopter-based radar data. Horizontal lines represent mean ± std over 50 m elevation bands).

illustrate the high spatial variability across the elevation range of each glacier and from one glacier to another across the basin (Figure 13). Elevation is the dominant influence on SWE at the Upper Susitna basin scale, with snow accumulation on the glaciers at 2000 m measuring 2-3 times higher than at 1000 m. A notable south-north decrease in total SWE and accumulation gradient indicates a strong orographic influence. Over short spatial scales in the ablation zone, surface roughness is responsible for high spatial variability in SWE. There is good correspondence between the radar measurements and the traditional method (Figure 13). A few points at high elevation where the discrepancies are largest might be explained by a misinterpretation of the location of the previous summer surface, either by manual measurement (probe) of a shallow ice layer, or by selection of a deeper firn layer in the radar data.

### 4.4 Continuous point mass balance measurements

Adjacent to the West Fork Glacier weather station (1398 m a.s.l.) we installed an acoustic distance sensor and a Wingscapes time lapse camera to measure snow accumulation and melt with high temporal resolution. The distance sensor was fixed vertically by mounting it on a pole drilled a few meters into the glacier, allowing the distance measured from the sensor to the





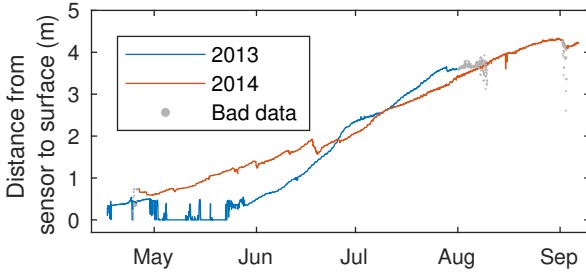

**Figure 14.** Distance from the acoustic sensor to the snow or ice surface. Increasing distance is a result of snow or ice melt and snow settling. Decreasing distance is due to snow accumulation. In early August 2013, the sensor's mounting pole began to tip over and give bad readings. On 1-2 September 2014, 18 cm of snow accumulation was recorded, consistent with observations during a site visit. The sensor pinged off falling snow, so some points in that window are labeled as bad data.

surface to be directly related to melt or accumulation. The time lapse camera was mounted to the On-Ice weather station and maintained a fixed height above the glacier surface. It took a picture facing the distance sensor every hour for the summer of 2013 and 2014.

Early in the 2013 record (16 April-28 May), the distance sensor was repeatedly covered by fresh snow accumulation, giving
inconsistent readings (Figure 14). From 28 May through 1 August 2013 the sensor gave reliable data and the measured net surface lowering due to surface mass balance was 3.3 m. After 1 August 2013, the sensor's mounting pole began to tip over and give meaningless readings. The tilt was observed in the time lapse imagery (sensor absent from 26 August 2013 image in Figure 15) . There was very little summer snowfall in 2013. There is a distinct increase in measurement noise after the transition from snow to ice. This is likely due to the roughness of the ice surface. The height change rate while the surface was
snow-covered (28 May - 25 June) was 0.053 m/day. The net depth of snow lost at the sonic distance sensor site was 1.97 m (17 April - 30 June), which is comparable to the 2.15 m snow depth measured in a snow pit about 6 m away on 14 April 2013. The ice surface lowered at a rate of 0.043 m/day (30 June - 1 August).

In 2014, the distance sensor gave good readings from 26 April to 1 September when the sensor was removed after a two-day snowstorm. The 2014 measured net surface lowering from the last significant spring snowfall (28 April) to the first fall snowfall
(1 September) was 3.8 m; summer snowstorms added 0.86 m of snow to the glacier which also melted away, for a total summer melt of 4.7 m. As in 2013, the 2014 data became noisier as the surface transitioned from snow to ice. The snow-covered surface lowered at a rate of 0.037 m d$^{-1}$ (28 April - 3 July) and the ice surface lowered at a rate of 0.030 m d$^{-1}$ (3 August - 1 September). Although the rate of surface lowering is larger for the snow surface than the ice surface, the rate of mass change is lower due to lower density.
To calculate ablation from the observed distance change, we assumed a density of 350 kg m$^{-3}$, based on snow density measurements at the site. This leads to an average melt rate of 0.016 m w.e. d$^{-1}$ for the summer of 2013 and 0.012 m w.e. d$^{-1}$ for 2014.




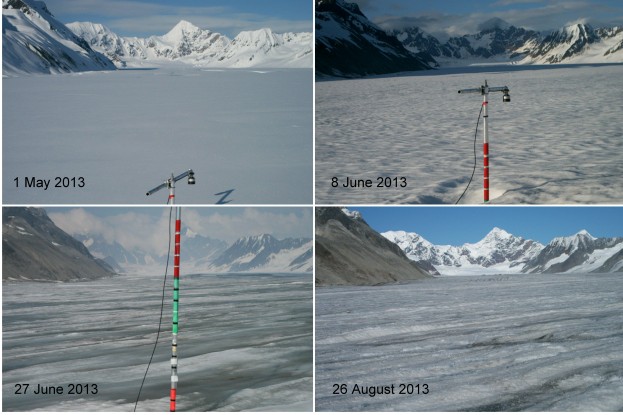

**Figure 15.** Photos from the time lapse camera on West Fork Glacier show surface conditions: Smooth fresh snow (1 May 2013), aged snow with rougher surface (8 June 2013), recently exposed bare ice with some water-saturated snow (27 June 2013), aged ice surface (26 August 2013).

## 5  Snow depth in non-glacierized terrain

### 5.1  Field measurements of snow in spring 2012 and 2014

In-situ snow measurements were made with a double sampling method (Rovansek et al., 1993) in non-glacierized settings (forest and tundra, see Figure 1 for locations). At each site, three to five snow cores capturing the entire snowpack were extracted and weighed to calculate snow density. In 2012 we used an Adirondak snow tube and in 2014 a SnowHydro snow tube. A larger number (∼50) of snow depth measurements were also taken to improve the statistical properties of the sample. Snow water equivalent (SWE) was calculated by multiplying the average snow depth with the average snow density. The sites were reached by helicopter (4 April 2012 and 22-28 April 2014) and via snow machine (8-12 April 2014). No measurements were obtained in 2013. The compiled measurements are presented in Table 6.

The field measurements showed variations in SWE according to elevation, region, and vegetation type but with the small number of sites, definitive statistics were not feasible. The SWE measurements illustrated a general increase with elevation in 2014, which became more significant in late April. Basin-wide SWE data distinguished three major regions (Maclaren, Clearwater and Talkeetna), where the Maclaren sites represented the highest SWE and the Talkeetna region the lowest. Within each region, the SWE data generally showed a strong elevation dependence. Among the two main vegetation types, shrubs presented larger SWE than the spruce locations (Table 6).

At the Lower Windy Cr. site, about 40 mm of SWE (25%) was lost due to melt of the end-of-winter snowpack between 9 April and 22 April 2014 (Table 6).





**Table 6.** Spring snow depth and density measurements in non-glacierized terrain, April 2012, early April 2014, and late April 2014. Snow depths were averaged from about 50 measurements in the vicinity of each site. Density were averaged from three to five sites.

| Site | Northing (m) | Easting (m) | Elevation (m a.s.l.) | Veg. type | Date | Snow depth (mm) | Depth $\sigma$ (mm) | Density (kg m$^{-3}$) | SWE (mm w.e.) |
|---|---|---|---|---|---|---|---|---|---|
| Early April 2012 | | | | | | | | | |
| Maclaren R. | 7003216 | 516825 | 941 | Shrub | April 04 | 1300 | 90 | 290 | 371 |
| Windy Cr. Upper | 6999971 | 492008 | 1185 | Shrub | April 04 | 880 | 370 | 280 | 248 |
| Windy Cr. Lower | 6998884 | 480035 | 939 | Shrub | April 04 | 760 | 160 | 270 | 202 |
| Open spruce forest | 6992239 | 478016 | 817 | Spruce | April 04 | 1050 | 100 | 130 | 141 |
| Early April 2014 | | | | | | | | | |
| Maclaren Glacier Lower | 7014353 | 523593 | 943 | Shrub | April 11 | 1430 | 150 | 310 | 440 |
| Maclaren Glacier Trail | 7001006 | 520732 | 996 | Shrub | April 11 | 1320 | 80 | 330 | 435 |
| 7mile Lake Trail Upper | 7003041 | 526340 | 1023 | Shrub | April 11 | 940 | 80 | 250 | 234 |
| 7mile Lake Trail Lower | 7000569 | 526655 | 965 | Shrub | April 11 | 790 | 100 | 250 | 197 |
| Top of Denali Hwy | 6995546 | 528779 | 1207 | Shrub | April 12 | 950 | 170 | 300 | 287 |
| Clearwater/Denali Hwy | 6990546 | 507510 | 942 | Shrub | April 11 | 860 | 70 | 240 | 202 |
| Clearwater Cr. | 6974167 | 493626 | 711 | Spruce | April 10 | 630 | 50 | 210 | 134 |
| Windy Cr. Upper | 6999971 | 491995 | 1198 | Shrub | April 09 | 1000 | 590 | 330 | 328 |
| Windy Cr. Lower | 6998893 | 480306 | 946 | Shrub | April 09 | 690 | 120 | 220 | 153 |
| Shrub/Tundra | 7009898 | 466122 | 911 | Shrub | April 08 | 510 | 140 | 280 | 141 |
| Spruce forest | 6992367 | 470690 | 781 | Spruce | April 09 | 680 | 100 | 220 | 150 |
| Open spruce forest | 6992232 | 478023 | 809 | Spruce | April 08 | 720 | 70 | 240 | 173 |
| Low sparse shrubs | 6986486 | 468969 | 931 | Shrub | April 09 | 1090 | 130 | 280 | 306 |
| Late April 2014 | | | | | | | | | |
| Two Plate Cr. | 7019108 | 518464 | 1555 | Rock | April 22 | 2590 | 230 | 270 | 687 |
| Maclaren Upper | 7003457 | 513239 | 1266 | Shrub | April 22 | 1760 | 230 | 350 | 610 |
| Maclaren Lower | 7004506 | 514754 | 1016 | Shrub | April 22 | 1310 | 80 | 270 | 348 |
| Windy Cr. Lower | 6998893 | 480306 | 946 | Shrub | April 22 | 540 | 170 | 210 | 114 |
| Tyone Cr. | 6903793 | 498152 | 954 | Spruce | April 28 | 110 | 100 | 270 | 30 |
| Oshetna Cr. Lower | 6901721 | 475712 | 1263 | Shrub | April 28 | 570 | 60 | 270 | 152 |
| Oshetna Cr. Upper | 6900723 | 458329 | 1583 | Shrub | April 28 | 430 | 140 | 270 | 113 |
| Kosina Cr. Upper | 6937020 | 451554 | 1274 | Shrub | April 28 | 320 | 280 | 230 | 72 |
| Kosina Cr. Lower | 6948876 | 450371 | 919 | Shrub | April 28 | 230 | 130 | 270 | 60 |

## 5.2 Snow depth in non-glacierized terrain: comparison to 1980s data

A total of 165 snow depth measurements, at 16 locations in both glacierized and non-glacierized terrain, were collected in 1981 and 1982 by R&M Consultants (1982). Direct comparisons to our data are hampered by differing measurement locations and dates, though the 1980s generally had larger snow depths.

## 6 Soils

### 6.1 Soil pit characterization

At each of 9 tundra sites, we dug a soil pit from the surface down to the top of the mineral soil between 1-3 October 2013 (see Figures 1, 16). We recorded the type of vegetation growing in the soil, visual characteristics of each soil horizon, as well as our estimation of the soil texture (Table A2). Two sites were too rocky to dig a pit (Two Plate Creek and Valdez Creek). The thinnest soils were generally observed at high elevation sites. The thickest organic-rich soils were observed in low-slope low-elevation environments at Tyone Creek and Maclaren Lower. Though we did not do a detailed texture analy-





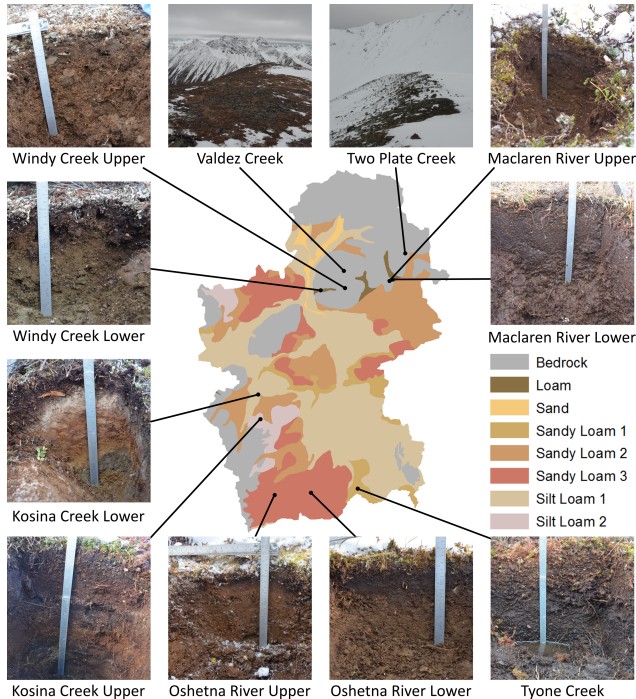

**Figure 16.** Photos of shallow soil pits. The map shows STATSGO soil type boundaries and pit locations. The ruler in each photo is 1.5 cm wide. Pit depths range from about 10 to 35 cm.

sis on the soils, the characteristics we observed generally matched up with the State Soil Geographic (STATSGO) soil map (https://datagateway.nrcs.usda.gov/).

## 6.2 Soil temperatures

Understanding the distribution of permafrost and seasonally-frozen ground across the basin is important for modeling of water

5   moving across the landscape. At each of the simple tundra weather stations installed over soil, we deployed soil temperature sensors at two depths below the surface (Figure 17). Shallow sensors (10 cm depth) were deployed into the side of the soil pits described above and then the pits were back-filled. Deep sensors were deployed at the bottom of a 2.5 cm-wide hole drilled into the ground. At some locations, the drill ran into rocks and was not able to reach the desired 1 m depth.

  Broad patterns that we observed in the data include: no permafrost in the upper layers of soil at the sites we sampled although

10   it may persist in deeper layers. The annual range of temperature at the shallow soil sensors is less than the air temperature range, and the day-to-day variability is damped too. The temperature cycles at the deep sensors (between 25 cm and 1 m depth, depending on the station) are even more damped than the shallow sensors.

  Soil temperature at three sites (Tyone Creek, Maclaren Lower, and Maclaren Upper) was nearly constant through the winter at or just above 0°C. At Tyone Creek and Maclaren Lower the ground surface was boggy and we observed liquid water seeping





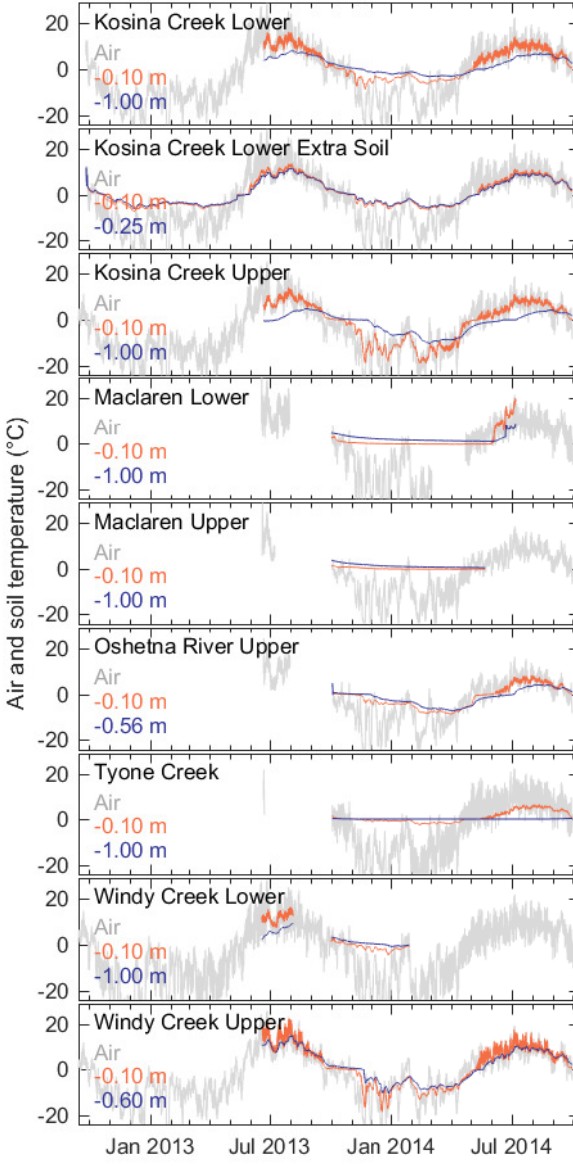

**Figure 17.** Hourly soil temperatures at depth below the soil surface for all sites with soils data, and air temperatures from each site's simple AWS. Each panel lists the depth of the soil temperature probes.

into our soil pits less than an hour after digging them (Figure 16), indicating that water flow through the upper unfrozen layer may have contributed to the nearly constant soil temperatures. Soils at Windy Creek Upper and Kosina Creek Upper get colder than the other stations in the winter due to the rocky soil and relatively high elevation of those two sites.





Figure 18 shows an example of summertime soil temperature fluctuations from the Windy Creek Upper station. Daily mean air and soil temperatures were all above 10°C. The shallow soil (0.1 m depth) at this rocky site heated up even more than the air each day, particularly when it was sunny. The diurnal cycle was smaller in the shallow soil than in the air and the deep soil (60 cm depth) showed only a very subtle diurnal cycle.

The middle panel of Figure 18 shows a soil freezing event from fall 2013 at Kosina Creek Upper. Both the shallow (0.1 m depth) and deep (1 m depth) sensors start the period at about 4 °C and show a cooling trend over the interval due to heat loss to the atmosphere (sub-zero air temperatures). For the first day of the period, the shallow sensor exhibited a diurnal cycle of temperature with a magnitude of 2 °C compared to diurnal amplitude in air temperature of 10 °C. In combination with the cold air temperatures, it is likely that this site received snow on 17 September 2013. The Gulkana Airport National Climatic Data Center (NCDC) station to the east received 8 and 12 mm w.e. of snow on 17 - 18 September. The Matanuska station to the south received 4 and 8 mm w.e. of snow on 18 - 19 September. Interestingly, Talkeetna Airport to the west did not receive snow during this period. The diurnal cycle of temperature disappeared by 18 September. The soil continued to cool throughout the interval, reaching 0.2 °C at 0.1 m depth and 1.1 °C at 1 m depth by 7 October 2013.

Spring thaw of the upper 0.1 m of soil at Kosina Creek Upper occurred over a period of about two weeks in 2014. On 20 April the shallow soil temperature sensor recorded a mean temperature of -1.9 °C and a diurnal cycle variation of 2°C while the air temperature diurnal amplitude was about 8 °C. As the air temperature warmed over the week of 27 April, the soil warmed to the freezing point and held steady until most of the soil had thawed. On 10 May, the shallow soil temperature exceeded 0°C for the first time that year and thereafter resumed a diurnal cycle. The 1 m deep soils warmed from -6.3 °C to -1.7 °C over this interval.

## 7   Data availability

Data is available at http://doi.org/10.14509/30138 (Bliss et al., 2019).

## 8   Conclusions

Comprehensive observations of meteorology, snow cover, glacier mass change, and soil properties are important for assessing basin-wide changes and providing input to hydrological modeling. In this study we focused on the Upper Susitna watershed - the source area that would feed a proposed hydroelectric dam. Our measurements reoccupied many of the same sites used by an initial study of the region 30 years ago. The 1980's measurements in combination with those presented here provide a baseline for future studies in the area.

Summer air temperatures in 2012-2014 were 1.1°C warmer than 1981-1983. Annual temperatures were 0.5°C warmer in the recent period. We found lapse rates to be significantly lower over glacierized surfaces in summer than over non-glaciated areas. Our meteorological stations filled a large gap in observations (spatially and elevation). Through correlations with long-running NOAA sites, we can better estimate past conditions within the basin.





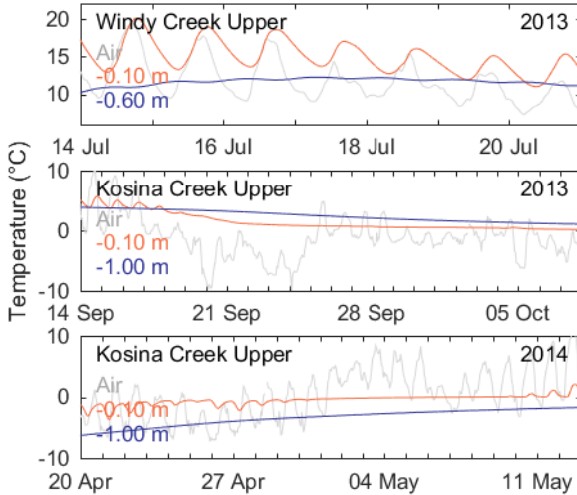

**Figure 18.** Hourly soil temperatures at depth below the soil surface illustrating three time periods: diurnal fluctuations in summer, soil freezing in fall, soil thaw in spring. Air temperature data is repeated from Figure 9 for context. Each panel lists the depth of the soil temperature probes.

Glacier surface mass balance measurements showed that during the melt season the glaciers were losing mass more than 3 times more rapidly in the 2010s than in the 1980s. Winter snow accumulation measured by traditional methods closely matched measurements gathered from a helicopter-borne snow radar. Annual glacier-wide mass balance went from being close to 0 m w.e. (balanced) in the 1980s to losing more than 1.5 m w.e. $yr^{-1}$ in recent years.

Snow depth in non-glacierized areas showed wide variability from site to site, reflecting complex deposition and redistribution patterns. Within local areas, higher elevations received more snow than lower elevations.

Our observations of soils in the basin generally match up with mapped soil descriptions. Soil temperature measurements revealed that none of the sites had permafrost in the upper 1 meter of soil. Most sites froze in the winter, though three sites remained at the freezing point despite air temperatures of -20°C.

The data sets described here provide new data in an extremely data scarce region. The data are valuable as baseline to assess future changes and will aid calibration and validation of hydrological, glaciological and other environmental models.

## 9 Author contributions

AB processed the data, performed all calculations and created all figures except Figure 1 (GW) and wrote most of the manuscript. RH, GW, and EW contributed significantly to the development of the analyses and figures, and the writing. AG

processed the raw radar data. All authors but WH and JZ contributed to the field work in the 2010s. WH provided information



on the 1980s data. CA and AH contributed to some initial data analyses. RH, GW, AL, and JZ secured funding from the Alaska Energy Authority.

## 10   Competing interests

The authors declare that they have no conflict of interest.

5   *Acknowledgements.*   The authors appreciate support and funding from: University of Alaska Fairbanks, Alaska Division of Geological and Geophysical Surveys, Alaska Energy Authority, and Colorado State University.




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

**Appendix A**



**Table A1.** Properties of meteorological stations referenced in this study. See figure 3 for example photos of our station types.

| Station name | Station type | Latitude | Longitude | Northing | Easting | Elevation |
|---|---|---|---|---|---|---|
| West Fork On-Ice ESG1 | AWS | 63.532 | -147.246 | 7044911 | 487787 | 1398 |
| Susitna Off-Ice ESG2 | AWS | 63.531 | -146.888 | 7044726 | 505559 | 1516 |
| Repeater ESR9 | AWS | 63.504 | -147.184 | 7041738 | 490839 | 2079 |
| EF1 | HOBO glacier | 63.436 | -146.596 | 7034211 | 520153 | 2133 |
| EF2 | HOBO glacier | 63.436 | -146.782 | 7034175 | 510895 | 1328 |
| EF3 | HOBO glacier | 63.413 | -146.811 | 7031586 | 509419 | 1073 |
| Mac1 | HOBO glacier | 63.416 | -146.599 | 7032005 | 520038 | 2104 |
| Mac2 | HOBO glacier | 63.355 | -146.550 | 7025220 | 522529 | 1396 |
| Mac3 | HOBO glacier | 63.305 | -146.527 | 7019700 | 523689 | 1018 |
| NWTrib1 | HOBO glacier | 63.598 | -146.986 | 7052170 | 500703 | 2075 |
| Off-Ice HOBO | HOBO glacier | 63.531 | -146.888 | 7044725 | 505561 | 1516 |
| Repeater HOBO | HOBO glacier | 63.504 | -147.184 | 7041736 | 490842 | 2079 |
| SU1 | HOBO glacier | 63.493 | -146.617 | 7040563 | 519096 | 1858 |
| SU3 | HOBO glacier | 63.512 | -146.887 | 7042599 | 505625 | 1245 |
| WF1 | HOBO glacier | 63.605 | -147.079 | 7053029 | 496062 | 1971 |
| WF5 | HOBO glacier | 63.487 | -147.449 | 7039879 | 477642 | 1123 |
| WFTranB | HOBO glacier | 63.529 | -147.237 | 7044519 | 488236 | 1413 |
| Kosina Creek Lower | HOBO tundra | 62.667 | -147.969 | 6948875 | 450373 | 919 |
| Kosina Creek Lower Extra Soil | HOBO tundra | 62.667 | -147.969 | 6948875 | 450373 | 919 |
| Kosina Creek Upper | HOBO tundra | 62.561 | -147.942 | 6937020 | 451553 | 1274 |
| Maclaren Lower | HOBO tundra | 63.170 | -146.707 | 7004512 | 514754 | 1016 |
| Maclaren Upper | HOBO tundra | 63.160 | -146.737 | 7003466 | 513262 | 1315 |
| Oshetna River Lower | HOBO tundra | 62.246 | -147.468 | 6901717 | 475705 | 1263 |
| Oshetna River Upper | HOBO tundra | 62.236 | -147.802 | 6900720 | 458336 | 1583 |
| Two Plate Creek | HOBO tundra | 63.300 | -146.632 | 7019112 | 518467 | 1555 |
| Tyone Creek | HOBO tundra | 62.266 | -147.036 | 6903805 | 498150 | 954 |
| Valdez Creek | HOBO tundra | 63.203 | -147.170 | 7008206 | 491438 | 1676 |
| Windy Creek Lower | HOBO tundra | 63.119 | -147.390 | 6998890 | 480303 | 941 |
| Windy Creek Upper | HOBO tundra | 63.129 | -147.159 | 6999977 | 491975 | 1177 |
| ALPINE CREEK LODGE AK US | NCDC | 63.043 | -147.248 | 6990393 | 487462 | 945 |
| GULKANA AIRPORT AK US | NCDC | 62.159 | -145.459 | 6892859 | 580297 | 476 |
| TALKEETNA AIRPORT AK US | NCDC | 62.320 | -150.095 | 6913666 | 339639 | 107 |



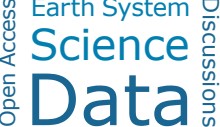

**Table A2.** Soil pit observations.

| Site Depth (cm) | Soil horizon | Description | Comments in the field |
|---|---|---|---|
| **Kosina Creek Lower (East face of pit)** | | | |
| -36 to 0 | Vegetation | | Willows, shrubs |
| -15 to 0 | Vegetation | | Crowberries, lowbush cranberries |
| -2 to 0 | Vegetation | | Green part of moss, lichens |
| 0 to 2 | Oi | Hemic | Moderately decomposed roots (MPM), dark brown color |
| 2 to 6 | 1 B | Mineral | Light buff color, 100% silt |
| 6 to 15 | 2 B | Mineral | Mottled coloration. Buff to reddish brown, 100% silt (slightly coarser than 1 B layer) |
| 15 to 18 | 3 B | Mineral | Yellowish brown color, 60% silt, 40% sand |
| 18 to 22 | 4 B | Mineral | Reddish brown color, 70% coarse grained sand, 30% silt |
| 22 | Bedrock | Bedrock | Granitic bedrock or boulder encountered |
| **Kosina Creek Lower (North face of pit)** | | | |
| -2 to 0 | Vegetation | | Green part of moss, lichens |
| 0 to 2 | 1 Oi | Moss | Brown part of moss |
| 2 to 10 | 2 Oi | Fibric | Slightly decomposed plant material (SPM) |
| 10 | Bedrock | Bedrock | Granitic bedrock or boulder encountered |
| **Kosina Creek Upper** | | | |
| -3 to 0 | Vegetation | | Lowbush cranberry, moss, lichen, occasional sedge tufts |
| 0 to 1 | 1 Oi | Moss | Brown part of moss |
| 1 to 6 | 2 Oi | Fibric | Slightly decomposed plant material (SPM) |
| 6 to 10 | Oe to Oa | Hemic to Sapric | Moderately to very decomposed roots (MPM transitioning to HPM) |
| 10 to 11 | 1 B | Mineral | Light brown, 80% silt, 20% sand, up to coarse-grained sand (quartz xl) |
| 11 to 16 | 2 B | Mineral ? | Dark brown color, predominately clay or HPM, 20% silt |
| 16 to 21 | 3 B | Mineral | Reddish light brown color. >90% fine-grained sand |
| 21 to 24 | 2 B | Mineral | Dark brown color, predominately clay or HPM, 20% silt |
| 24 to 32 | 4 B | Mineral | Light brown, 60% medium-grained sand, 30% silt |
| **Maclaren River Lower** | | | |
| -20 to 0 | Vegetation | | Sedges, grasses, arctic cotton plants |
| -1 to 0 | Vegetation | | Green part of moss, lichens |
| 0 to 1 | 1 Oi | Moss | Brown part of moss |
| 1 to 4 | 2 Oi | Fibric | Slightly decomposed plant material (SPM) |
| 4 to 35 | Oe | Hemic | Moderately decomposed roots (MPM), could not find mineral horizon |
| 35 | Base of pit | | |
| **Maclaren River Upper** | | | |
| -6 to 0 | Vegetation | | Willows, crow berry, lowbush cranberry, some moss |
| 0 to 2 | Oi | Fibric | Slightly decomposed plant material (SPM) |
| 2 to 11 | Oe | Hemic | Moderately decomposed roots (MPM) |
| 11 to 20 | B | Mineral | 40% silt, 20% sand, 40% clay (estimated percentages) |
| 20 | Base of pit | | |
| **Oshetna River Lower** | | | |
| -10 to 0 | Vegetation | | Grass, sedges |
| -2 to 0 | Vegetation | | Green part of moss, lichens |
| 0 to 1 | 1 Oi | Moss | Brown part of moss |
| 1 to 5 | Oa | Sapric | Highly decomposed plant material (HPM), dark brown soil color |
| 5 to 13 | B | Mineral | Medium brown soil, 5% reddish medium brown spots, 90% silt, <10% clay, occasional gravel (1 -3 cm diameter) |
| 13 | Base of pit | | |
| **Oshetna River Upper** | | | |
| -8 to 0 | Vegetation | | Shrubs |
| -2 to 0 | Vegetation | | Green part of moss, lichens |
| 0 to 1 | 1 Oi | Moss, Fibric | Brown part of moss + SPM |
| 1 to 2 | Oe | Hemic | Moderately decomposed plant material (MPM) |
| 2 to 17 | B | Mineral | Moderate brown color, clay % > silt % > gravel % (2-3 cm diameter) |
| 17 | Base of pit | | |
| **Tyone Creek** | | | |
| -10 to 0 | Vegetation | | Sedges, shrubby pine up to 20 cm tall in site vicinity |
| -4 to 0 | Vegetation | | Blueberries, crow berries |
| -3 to 0 | Vegetation | | Green part of moss |
| 0 to 2 | 1 Oi | Moss | Brown part of moss |
| 2 to 7 | 2 Oi | Fibric | Slightly decomposed plant material (SPM), light brown soil color |
| 7 to 27 | Oe | Hemic | Moderately decomposed plant material (MPM), small % of live shrubby pine roots |
| 27 | Base of pit | | |
| **Windy Creek Lower** | | | |
| -2 to 0 | Vegetation | | Lichen, lowbush cranberry, willows |
| 0 to 3 | Oi | Fibric | Slightly decomposed plant material (SPM) |
| 3 to 5 | Oe to Oa | Hemic to Sapric | Moderately to very decomposed roots (MPM transitioning to HPM) |
| 5 to 15 | B | Mineral layer | Sandy % > silt % |
| 15 | Base of pit | | |
| **Windy Creek Upper** | | | |
| -1 to 0 | Vegetation | | Lichen, lowbush cranberry |
| 0 to 6 | ? | Thin, sparse roots | Medium brown soil, not much organic material, 70% silt, 30% clay, gravel (1 - 5 cm) |
| 6 to 17 | B | Rocky | Medium light brown soil, 70% silt, 30% clay |
| 17 | Base of pit | | |