# Peer review of "Glaciers and Climate of the Upper Susitna Basin, Alaska"

_Earth System Science Data, 2018_

## Referee Comment (RC1) · Anonymous Referee #1 · 25 Jun 2019

General comments: This paper presents a dataset in the Upper Susitna Basin in Alaska and contains the data itself, the data collection and data processing of meteorological, glacier mass balance, snow cover and soil measurements. Data collection in this region is hard and this data can be great value for model validation or calibration in this region, however, I think not in the way it is framed now by focussing on implications for the dam (that is not there) and climate change (data is too limited).

I think the dataset presentation can be more thorough and better structured than is presented now. I miss at various places context of statements and the naming of the different stations can be structured better, such that it is clear to the reader which data you are referring to. I specified this in the specific comments below.

I suggest major comments, predominantly since I think the focus of this paper should

[Figure]

be on the data and not too strongly framed to dam-implication work and/or climate change work, since this work does not address that.

Specific comments: The introduction does not focus on the relevant subjects. You present solely a measurement dataset and you focus in the introduction on climate change and (modelled) river runoff, which you did not do in the paper. Please restructure the introduction, remove this information or at least shift focus to the data. You could include more information about previous field works/data sets in this region?

In general: update the captions of figures, such that those are complete. In Figure 1 I miss for example explanation of the subpanels. Also update the figure labels and text inside the figures such that those are readable (mainly Figure1) and resolution is high enough (also for the tables).

P1L16: You did not raise any questions yet. Please rephrase. P1L16: Your introduction is focused on climate change and river runoff, however I do not think 1 data set is sufficient to solve that. I would focus more on fundamental understanding rather than climate change. Please shift the focus of the importance of the data or even present it only as a dataset.

P2L1-4: in the paper itself you do not make the link to water runoff or the dam so I find this information misleading. P2L5: state 127 glaciers instead of 'more than 120'. Be specific P2L9-10 similar as stated above, you do not make this link to dam operations and environmental resources. Or looked at river discharge. P2L11-14: So you do not include modelling (as stated in P2L3)?

Section 2. Study area: A lot of unnecessary information. Please exclude some information or make more clear why you discuss this information. Connect the information to your paper and rewrite to a smoother story. P2L19-21 "modern glaciers. . . Ice sheet" Surging Permafrost

P2L19: why Kienholz reference needed? P2L24: "few"=how many glaciers? P2L28:

change "ninety-three" → 93 P2L28: "127 glaciers": numbers are not congruent ("more than 120 glaciers" in L5, "Most glaciers (in total 127)" in L23). If the area contains 127 glaciers please change in L5 to "more than 120 glaciers" to "127 glaciers", remove in L23 "(in total 127)". P2L29: how did you estimate this volume? Explain in text. P2L30: why do you use these scaling coefficients? Please explain how you got those or insert reference. P2L31. Place '.' After m a.s.l.. P2L33: change point into comma or insert 'and'

Page 3: Figure1: Combine Figure 1 and 2 by shading/colouring the permafrost areas >50% in Figure 1. For reader it is not clear why permafrost is so important. P3L1: Why is debris cover relevant? P3L1-2 please rephrase and combine L1 and 2 P3L5: include Östrom reference. P3L6: Do you exactly study the same area? P3L10: "History of surging", are they still defined as surging glaciers or long time stable aleardy??

P4L1-8: why is surging relevant? Mass balance not tat different than non-surging glaciers. P4L9: this line suggests that glacierized parts have vegetation and human development. Please rephrase. P4L9-15: why is permafrost relevant?

Page 5 Figure 2: Do not understand the importance of this figure, maybe only the two upper permafrost colors needed? Than combine with Figure 1.

Page 5 Figure 3: put all photos next to each other or make panel a and d bigger such that they are similar height as b+c.

P6L4: do you refer to figure three, or do you mean you installed three AWS? Are those the 'station type AWS' in Table A1? Not clear for reader that energy balance weather stations are 'AWS' and that simple AWS is 'HOBO' in table A1. P6L6: "floated" is not appropriate here. P6L25: "The station records…surface elevation changes", put earlier in section (for example at the end of P6L4.

Page 7: Figure 4: caption "(i) outgoing longwave radiation" use abbreviation LW as defined before.

Page 8: Table 1: It is not clear to me whether the "energy balance weather stations" are the onces indicated by "AWS" of "HOBO" in Table A1? Please use consistent naming for all stations throughout paper. P8L1-2 : what do you want to show with correlation coefficients in Table 2? P8L4: "variable", you indicate only the range of the data in Table 2. Please provide also the standard deviation of the values in Table 2 and refer to the standard deviations in the text. P8L7: "more than 4 degrees Celcius lower": please rephrase to "temperatures by a minimal bias of 4 degrees" or similar. P8L8: connect the sentences by adding "when" P8L6-10: structure is missing, please rephrase P8L11-13: please provide numbers instead of only calling it 'higher', 'greater availability', 'less variability' etc.

P9 Table 2: daily correlations? Add standard deviations to the means to indicate the variability. You can remove the range, since that does not add more information than the standard deviation

P10L9: Incoming longwave radiation is also influenced by the surroundings, especially in complex terrain. Insert "mainly"→"which depends mainly on the effective.." P10L12: high correlations means meteorology is affected by larger scale forcings rather than micrometeorology. You could add discussion about that. P10L23: change "Figure 5" to "Figure 5C". and add ABC in Figure 5 P10L25: not necessary melt when T>0 degress, the surface energy balance should be positive. P10L30: What about the influence of precipitation. P10L31: rephrase "in the ice rather than the snowpack" to "the layer consists of ice instead of snow" or similar

P11 Figure 5: explain the reversed patter of ice temperature (red lines) with depth in the text. At 21 April lines are ordered from light to dark lines with depth, while in June this pattern is reversed. With other words explain why the temperature gradient reverses. the lightest lines (Ice2.5m) are not clearly visible and Ice3m not present at all. Please make those lines more clear.

P12 Figure6: the reversed temperature gradient is here not visible why (not)?  Go

more in depth in the data (general comment) P12L4: "simple weather stations"? Why is station type in Table A1 than indicated by "HOBO glacier"? please make naming consistent throughout the paper.

P13L7: which of the two sensors are more trustworthy? And why the comparison? Please explain in text. P13L9-11: I think these argumentations do not match: The HOBO sensor is slower than the Campbell, but coefficient of 1, and then conclusion is that there is a lack of consistent pattern. I do not follow this, please explain and rephrase P13L20: add some explanation/conclusion. I miss in this whole section why you do the comparison between the sensors and eventually the physical interpretation or conclusion from your statements.

P14L1-2: did more people had this problem? Is it a random tip that can also occur during dry periods (since this can not be filtered out)? Or is the tip sometimes 'stuck'? P14L4: or conclusion is the HOBO has a sensor problem. P14L16: is this katabatic flow measured or a assumption it develops?

P15 Figure 7: include the colours in the caption

P16-P17: Section 3.3 I do not think this section is a great addition to your purpose of the paper and not supported by any in depth discussion, please remove.

P18 Figure 11: is this the same transect measured every week at same location? What do you mean with "plotted relative to a reference station"? Does this mean steepness in line is varying in time? Please explain in caption. Mention in caption what upper stations in winter are not operating/measurement problems.

P19 figure 12: Add coloured lines for each of the dots to show whether the gradients change in time/how sensitive they are. Add the resolution of the glacier inventory in the caption. Insert in caption how the mass balance in computed (from the "HOBOglacier" station in Table A1?

P19L1-2: or the measurements are not representative for the whole region. P19L4-

end: the linear interpolation is done with all data such that no division is made between years (1 average value for all glaciers and all times?)? If so this is a very simplified method and I am not convinced in the numbers you present. For example P is highly spatially variable, as you also state in P20L14-15.

P20 Section 4.2 Very limited, please expand or consider removing or merging this section. P20L17: I do not follow this, you did not do any model simulation of snow accumulation, or this is not mentioned. P21L2-3 please rephrase. Absolute difference at 2000 and 1000m or did you do some averaging? P21L4: how do you know surface roughness is responsible? Please add explanation or supporting material for this statement. P21L6-8 I am not convinced

P22L9: again, why the roughness of the ice surface? P22L16: add explanation why data become noisier. Why does ice give more noise signal? P22L20: you assumed constant density? What are the implications of this assumption?

P24 Table 6: increase fontsize Section 5.2: please remove, I do not think is Section is of additional value P24L11: "though we did not do a detailed texture analysis", But still you know it matches with the STATSGO soil map? I am not convinced.

P25 Section 6.2 Explain the uncertainties and effect by the disturbance of the disturbed soil on the measurements.

P27L11-13: why is this relevant, please remove. P27L25: How do you relate this to the dam? No runoff analysis is done. P27L27: this is only a minor section and for me not strong part of your paper and now you present the climate change numbers as one of your main conclusions.

P28 Figure 18: air temperature is gray colour? P28L6: not new conclusion, snow amounts are generally higher at high elevations P28L7: you did not measure the soil and your conclusion is that these match with the mapped soil descriptions. Please remove this statement out of your conclusions and preferably also out of the text

Conclusion in general: please do not focus on climate change and dam implications, but give conclusions about the data you found in the field. What did you find and why is it special?

---

## Referee Comment (RC2) · Anonymous Referee #2 · 2 Jul 2019

General Comments:

This study presents, validates and interprets a comprehensive and impressive data set, which covers a range of parameters in the variable environments of the Susitna Basin, Alaska. The data set includes meteorological, glaciological and soil parameters. The data set is unique, as many of the measurements were done in complex terrain where measurements generally are sparse. It is effortful and requires extensive planning to acquire meaningful data in this terrain. Problems in the data are addressed and generally, implications that arise with these problems are described in detail. Overall, the manuscript is well structured and provides a good overview of the data. The data set itself could be extremely valuable for model validation or comparison with future field studies. However, the manuscript is not always coherent and suffers from redundant

information (e.g. section 3.3, section 5.2, figure 9), which distracts the reader from following the key points and weakens the focus of the paper (see specific comments). The introduction does not adequately motivate the manuscript, as it does not really make clear why the data set is important and what the purpose of the data set is (see specific comments). In addition, some of the presented data appear isolated and need to be put into context better (e.g. section 4.4., how do continuous mass balance measurements compare to stake measurements nearby?) Therefore, I suggest a number of minor revisions to focus the main messages of the manuscript and to emphasize the uniqueness of the data set.

Specific Comments:

Introduction: Mentioning climate change in the beginning of the introduction is not convincing, since you only acquire three years of data. Either remove the link to climate change, or emphasize that the data set is meant to be used for comparison with future studies (as you do in the conclusions). P2, l9-10: using changes in river flow on dam operations as a motivation here seems misplaced, since you mention in the beginning that the dam was not built. In addition, river flow is not covered in this study. Please rephrase or remove. You could motivate each of the data types (meteorological/climatological, glaciological, snow, soil) individually, as you do later in the manuscript. E.g., p4,l23-26 motivates meteorological/climatological measurements, p25,4-5 motivates soil measurements and should be placed in the introduction

- p1, l6: since only the years 1981-83 were investigated, please do not write 1980s here but refer specifically to the years 1981-83

- p2, l5: state precise number instead of "more than 120 glaciers"

- p3, l5: please add reference here

- p3, l10-p4,l3: detailed description of surge history seems unnecessary here, please shorten

- p6, l4: what does the number 3 in brackets mean?

- p6, l26: first time abbreviation "DGGS" is mentioned, please provide full name

- p7, figure 4: please provide figure in higher resolution

- p8, l13: delete sentence "Data are not available from 18 January 2014 to 22 April 2014 when the station was buried in snow." since this was mentioned before.

- p9, table 2 caption: delete "18 January 2014 to 22 April 2014." from caption, since it was mentioned before

- p11, figure 5: please provide higher resolution figure

- p12, figure 6: figure is not mentioned in text, please add reference to figure in text. Also, please provide figure in higher resolution

- p13, l8: reference to figure 4: is this right or do you mean figure 7?

- p13, l11: rather say "very close to 1.0"

- p13, l12/13: "The lack of a consistent pattern in these comparisons prevented us from adjusting the HOBO temperature data to match the Campbell data." This is confusing since you mention an average offset the sentence before. Can you clarify this?

- p.14, l1/2: What is your confidence that no double tips are missed or that normal tips are identified as double tips?

- p15, figure 7: how do you explain very high RH offset of some HOBO-sensor at higher RH, especially in 2013?

- p14, l11: "Precipitation amounts did not correlate significantly with elevation, slope, aspect, or location." How do you define "location"? What drives the variations in precipitation amounts?

- p14, l15-17: katabatic wind flow: Can you back this with references or add a more thorough analysis based on your data, e.g. wind direction analyses? Or is this just an

assumption you make?

- p15, figure 7: colour codes are missing

- p16, figure 9: what is the purpose of figure 9? There is no in-depth analysis provided in the text and patterns are trivial (yearly temperature cycle, lower temperatures with higher elevation). In addition, it figure again suffers from relatively low resolution. Please either remove figure or provide more detailed analysis

- p16, section 3.3: this section does not provide a thorough analysis and is not useful for the manuscript since it is not based on your data. Either please remove or transform; rather than a trend analysis, the section could provide an assessment whether the years 2013-2014 were exceptional (in terms of temperature) or normal.

- p18, figure 11: what purpose do the different colours serve? If none, please use black or dark grey

- p19, l12: Did you think about adding the line fit to the figure? This would allow the reader to clearly identify the equilibrium line altitude you derived

- p19, l1/2: Since you used only point measurements, it is also possible that the stake measurements do not fully represent the area that was covered by satellite. In addition, you mentioned earlier that most of the stakes were placed on the centreline of the glacier, which is typically higher than the margins (which are included in the satellite estimation?), potentially leading to higher estimation of the equilibrium line altitude

- p19, l7/8: "Glacier-wide mass balance estimates were then calculated by summing the distributed mass balance over the whole glacier." Did you use hypsometry of each individual glacier? Or did you use hypsometry of the entire glacier area for the calculation of the individual glacier mass balances? If so, the numbers you get probably have very high uncertainties. Please clarify.

- p20, l2: "East Fork Glacier had a similar mass balance as Maclaren Glacier." Why do you stress this here? Seems misplaced, please delete
- p20, l17: "To robustly validate model simulations of snow accumulation. . ."; please move to introduction, since this provides a motivation for your measurements and you are presenting results in this section

- p21, l2/3: ". . .at 2000 m measuring 2-3 times higher than at 1000 m." looks more like 2-4 times higher (see year 2014)

- p21, l4: "A notable south-north decrease in total SWE and accumulation gradient indicates a strong orographic influence." Please remove: sentence is redundant since you mention elevation dependence the sentence before. In addition, this is not always necessarily a north-south gradient

- p22, figure 14 caption: "In early August 2013, the sensor's mounting pole began to tip over and give bad readings. On 1-2 September 2014, 18 cm of snow accumulation was recorded, consistent with observations during a site visit. The sensor pinged off falling snow, so some points in that window are labeled as bad data." please remove or move to text

- p22, l16: "data became noisier as the surface transitioned from snow to ice." can this be seen in figure 14?

- p22, l21/22: "This leads to an average melt rate of 0.016 m w.e. d−1 for the summer of 2013 and 0.012 m w.e. d−1 for 2014." This information seems a bit isolated from the previous, interesting mass balance investigations. Can you provide a comparison here? How does this compare to nearby ablation stakes summer mass balances?

- p23, l7: "Snow water equivalent (SWE)": abbreviation has been initialized before

- p23, l14: ". . .generally showed a strong elevation dependence." dependence of what type? Maybe just write "increased with elevation"

- p23, l16/17: "At the Lower Windy Cr. site, about 40 mm of SWE (25%) was lost due to melt of the end-of-winter snowpack between 9 April and 22 April 2014 (Table 6)" Why is this stressed here? Please remove

- p24, section 5.2: This section does not add any value to the paper but is very distracting; please remove

- p25, l1: "characteristics we observed"; please specify these characteristics so the agreement between soil pits and STATSGO becomes clearer

- p25, l4/5: "Understanding the distribution of permafrost and seasonally-frozen ground across the basin is important for modeling of water moving across the landscape"; again, this provides a motivation for your measurements and you are presenting results in this section, so please move to introduction

- p27, l28/29: "Summer air temperatures in 2012-2014 were 1.1âŲęC warmer than 1981-1983. Annual temperatures were 0.5°C warmer in the recent period." Why do you add this here? This is not based on your own measurements and thus not a significant outcome. Please consider removing.

- p28, l2: since only the years 2012-14 vs. 1981-83 were investigated, please do not generally say 2010s vs. 1980s since you have no information on the other years

Technical Corrections:

- p8, l12: move "On-Ice" to the end of this sentence

- p6, l33: please remove "instead of every minute"

- p16, l4: "most distinct" instead of "least complicated"

- p18, l2: "refers to the period October..." instead of "refers to October..."

- p20, l11/12: "Therefore, lower annual balance in the latter period (-1.72±0.87 m w.e.) compared to the former period (0.04±0.25 m w.e.) were driven by the more negative summer balances." should be "balances... were driven..." or "balance... was driven"

- p26, figure 17 caption: "soil" instead of "soils"; remove "," after data

---

## Author Comment (AC1) · 9 Oct 2019

**RC1:**

General comments:
This paper presents a dataset in the Upper Susitna Basin in Alaska and contains the data itself, the data collection and data processing of meteorological, glacier mass balance, snow cover and soil measurements. Data collection in this region is hard and this data can be great value for model validation or calibration in this region, however, I think not in the way it is framed now by focussing on implications for the dam (that is not there) and climate change (data is too limited).
*We've made a set of changes to address the framing (see below), but the dam proposal was the main motivation for the project (and the reason the project was funded) so we chose to leave this mention in the intro. The wording makes it clear that the paper focuses on the data, this section of the intro is providing the context for why the data was collected. We added phrases about data scarcity and model validation and calibration to bolster the motivations.*

I think the dataset presentation can be more thorough and better structured than is presented now. I miss at various places context of statements and the naming of the different stations can be structured better, such that it is clear to the reader which data you are referring to. I specified this in the specific comments below.
*Addressed specifics below*

I suggest major comments, predominantly since I think the focus of this paper should be on the data and not too strongly framed to dam-implication work and/or climate change work, since this work does not address that.

Specific comments:
The introduction does not focus on the relevant subjects. You present solely a measurement dataset and you focus in the introduction on climate change and (modelled) river runoff, which you did not do in the paper. Please restructure the introduction, remove this information or at least shift focus to the data. You could include more information about previous field works/data sets in this region?
*We believe this larger context will be important to many readers. For people unfamiliar with the area, this gives them enough information that they don't have to search elsewhere for an understanding of why our data might be interesting or useful.*

In general: update the captions of figures, such that those are complete. In Figure 1 I miss for example explanation of the subpanels. Also update the figure labels and text inside the figures such that those are readable (mainly Figure1) and resolution is high enough (also for the tables).
*Added to caption of figure 1: "The main map focuses on the glacierized portion of the basin, the large inset shows the whole Upper Susitna basin which drains to the proposed dam site, and the small inset shows the basin in the context of the state of Alaska."*

P1L16: You did not raise any questions yet. Please rephrase.
*Done*

P1L16: Your introduction is focused on climate change and river runoff, however I do not think 1 data set is sufficient to solve that. I would focus more on fundamental understanding rather than climate change. Please shift the focus of the importance of the data or even present it only as a dataset.
*It is true that this data set won't "solve" climate change or changes to river runoff, but the introduction is meant to connect the paper to the larger topics that it relates to. That's what we're doing here. The wording makes it clear that the paper focuses on the data, this section of the intro is providing the context for why the data was collected. We added a sentence about data scarcity to bolster the motivations.*

P2L1-4: in the paper itself you do not make the link to water runoff or the dam so I find this information misleading.
*We include this information for context and motivation. On line 4 we clearly state "this paper focuses on the measurements."*

P2L5: state 127 glaciers instead of 'more than 120'. Be specific
*While we are generally in favor of being specific, here we use 'more than 120' because the number of glaciers in the basin can change e.g. as climate change removes glaciers, or splits one into two. I changed it to a nice round 'more than 100.'*

P2L9-10 similar as stated above, you do not make this link to dam operations and environmental resources. Or looked at river discharge.
*We include this information for context and motivation. On line 4 we clearly state "this paper focuses on the measurements."*

P2L11-14: So you do not include modelling (as stated in P2L3)?
*We include the information on modeling for context and motivation. On line 4 we clearly state "this paper focuses on the measurements."*

Section 2. Study area: A lot of unnecessary information. Please exclude some information or make more clear why you discuss this information. Connect the information to your paper and rewrite to a smoother story.
Ice sheet
Surging
Permafrost
*We believe all the information presented is useful background to introduce the reader to the area we're focusing on. In fact we already connect the information to our paper: "The glacier monitoring work focused on the five largest glaciers..." We include a succinct summary of the characteristics of the glaciers (area, debris cover, area change, surging behavior, and velocities) to give the reader a sense of the glaciers' behavior and previous work that has been completed on the glaciers. The one sentence on ice sheet extent provides context. The paragraph on surging is important because surging can dramatically influence the mass balance of a glacier (one of the main foci of this paper). The two sentences on permafrost and the map (figure 2) provide context for the soil temperature measurements.*

 P2L19: why Kienholz reference needed?
*Because Kienholz and co-authors did the work we're building on here. Their outlines were then incorporated into the RGI.*

P2L24: "few"=how many glaciers?
*Nine, as mentioned on line 32.*

P2L28: change "ninety-three" → 93
*Convention states that sentences should not start with Arabic numerals.*

P2L28: "127 glaciers": numbers are not congruent ("more than 120 glaciers" in L5, "Most glaciers (in total 127)" in L23). If the area contains 127 glaciers please change in L5 to "more than 120 glaciers" to "127 glaciers", remove in L23 "(in total 127)".
*127 in Alaska Range + 9 in Talkeetna Mountains, but see also my reply to P2L5*

P2L29: how did you estimate this volume? Explain in text.
*The first part of this sentence explains how we estimate volume.*

P2L30: why do you use these scaling coefficients? Please explain how you got those or insert reference.
*Added:*
*Radic, V., Hock, R., and Oerlemans, J.: Analysis of scaling methods in deriving future volume evolutions of valley glaciers, Journal of Glaciology, 54, 601–612, https://doi.org/10.3189/002214308786570809, http://dx.doi.org/10.3189/002214308786570809, 2008.*

P2L31. Place '.' After m a.s.l..
*Done*

P2L33: change point into comma or insert 'and'
*That is a period at the end of the sentence.*

Page 3: Figure1: Combine Figure 1 and 2 by shading/colouring the permafrost areas >50% in Figure 1. For reader it is not clear why permafrost is so important.
*Figure 1 is too complicated to add the permafrost classes and keep it readable. I added text to explain significance. "Permafrost affects water runoff, soil temperature, vegetation, and soil carbon fluxes. These factors have complex interactions with climate change."*

P3L1: Why is debris cover relevant?
*P3L4 explains why. I reordered these sentences.*

P3L1-2 please rephrase and combine L1 and 2
*No longer needed with reordering.*

P3L5: include Östrom reference.
*Upon searching, I can't find any relevant reference by that author.*

P3L6: Do you exactly study the same area?
*Yes, which is why we felt we did not need to say that we study the same area.*

P3L10: "History of surging", are they still defined as surging glaciers or long time stable aleardy??
*That's an interesting debate, that we choose not to delve into here.*

P4L1-8: why is surging relevant? Mass balance not tat different than non-surging glaciers.
*Mass balance can in fact be affected by surging (and vice versa). The latest surge of West Fork occurred shortly after the 1981-83 mass balance measurements to which we are comparing our data. While the effect of a surge on the mass balance may not be quantifiable given the data that is available, the surge still needs to be mentioned.*

P4L9: this line suggests that glacierized parts have vegetation and human development. Please rephrase.
*Text reads "The non-glacierized part..."*

P4L9-15: why is permafrost relevant?
*Added text to explain significance. "Permafrost affects water runoff, soil temperature, vegetation, and soil carbon fluxes. These factors have complex interactions with climate change."*

Page 5 Figure 2: Do not understand the importance of this figure, maybe only the two upper permafrost colors needed? Than combine with Figure 1.
*See above*

Page 5 Figure 3: put all photos next to each other or make panel a and d bigger such that they are similar height as b+c.
*Figure is designed to be one-column wide and be aesthetically pleasing with the mix of vertical and landscape orientations.*

P6L4: do you refer to figure three, or do you mean you installed three AWS? Are those the 'station type AWS" in Table A1? Not clear for reader that energy balance weather stations are 'AWS' and that simple AWS is 'HOBO' in table A1.
*Added "Figure" to the 3. Edited appendix to remove "AWS" and "HOBO".*

P6L6: "floated" is not appropriate here.
*Floated is the best descriptive word we have for this installation type. Added quotes around it to make clear that it is not floating in the literal sense.*

P6L25: "The station records…surface elevation changes", put earlier in section (for example at the end of P6L4.
*We mention this sensor on line 14, in what we think is a logical place to mention it.*

Page 7: Figure 4: caption "(i) outgoing longwave radiation" use abbreviation LW as defined before.
*Done*

Page 8: Table 1: It is not clear to me whether the "energy balance weather stations" are the onces indicated by "AWS" of "HOBO" in Table A1? Please use consistent naming for all stations throughout paper.
*Edited appendix to remove "AWS" and "HOBO".*

P8L1-2 : what do you want to show with correlation coefficients in Table 2?
*Section 3.1.2 describes what we want to show with the correlations.*
P8L4: "variable", you indicate only the range of the data in Table 2. Please provide also the standard deviation of the values in Table 2 and refer to the standard deviations in the text.
*The standard deviations are not particularly meaningful given the short records we're dealing with, so we chose not to include them. The reader can get a sense of the variability we refer to here by looking at figure 4.*

P8L7: "more than 4 degrees Celcius lower": please rephrase to "temperatures by a minimal bias of 4 degrees" or similar.
*Replaced "lower" with "cooler"*

P8L8: connect the sentences by adding "when" P8L6-10: structure is missing, please rephrase
*Reworded.*

P8L11-13: please provide numbers instead of only calling it 'higher', 'greater availability', 'less variability' etc.
*The table and figure have the numbers. Adding all the numbers here makes the text unreadable.*

P9 Table 2: daily correlations? Add standard deviations to the means to indicate the variability. You can remove the range, since that does not add more information than the standard deviation
*Standard deviation is not meaningful for wind speed, precipitation, or shortwave radiation since their distributions are so skewed. Range does give important information. The reader can see the variability in the figure.*

P10L9: Incoming longwave radiation is also influenced by the surroundings, especially in complex terrain. Insert "mainly"→"which depends mainly on the effective.."
*Done*

P10L12: high correlations means meteorology is affected by larger scale forcings rather than micrometeorology. You could add discussion about that.
*We can't generalize from "LW" to "meteorology." P10L2 already discusses the "relatively homogeneous cloud conditions at those sites"*

P10L23: change "Figure 5" to "Figure 5C". and add ABC in Figure 5
*Done*

P10L25: not necessary melt when T>0 degress, the surface energy balance should be positive.
*We are not directly measuring sensible or latent heat fluxes, so we can't calculate the surface energy balance precisely.*

P10L30: What about the influence of precipitation.
*There was no precipitation on these days, which is why we don't mention it.*

P10L31: rephrase "in the ice rather than the snowpack" to "the layer consists of ice instead of snow" or similar
*Done*

P11 Figure 5: explain the reversed patter of ice temperature (red lines) with depth in the text. At 21 April lines are ordered from light to dark lines with depth, while in June this pattern is reversed. With other words explain why the temperature gradient reverses.
*"As air temperatures rose, the subsurface temperatures of the upper layers increased but with a time lag that increased with 25 depth (Figures 5 and 6)."*

P11 Figure 5: the lightest lines (Ice2.5m) are not clearly visible and Ice3m not present at all. Please make those lines more clear.
*With 15 lines we did our best to have distinct colors that show the short term variability. Perhaps the confusion comes because we do not have data from 4 m depth so there is a visual gap between the data from 3 m and 5 m, particularly at the beginning of the record. The legend clearly indicates the depths.*

P12 Figure6: the reversed temperature gradient is here not visible why (not)? Go more in depth in the data (general comment)
*The reversed temperature gradient is visible. On 28 April the coldest temperatures are at the surface. On 26 May the coldest temperatures are at -1 m from the initial snow-ice interface, with warmer temperatures at the surface. The pattern continues through the end of the season.*

P12L4: "simple weather stations"? Why is station type in Table A1 than indicated by "HOBO glacier"? please make naming consistent throughout the paper.
*Edited appendix to remove "AWS" and "HOBO" in favor of "energy balance" "simple glacier" and "simple tundra."*

P13L7: which of the two sensors are more trustworthy? And why the comparison? Please explain in text.
*The manufacturer's stated accuracies are 0.1 C for Rotronic and 0.2 C for HOBO as shown in Tables 1 and 3. Whether that equates to "trustworthiness" is a matter of opinion. The comparison is done for calibration (as stated in the section header). Added a new first sentence: "To ensure the validity of data collected from many individual sensors scattered across our study area, we set them up side-by-side before deploying them to the field."*

P13L9-11: I think these argumentations do not match:
The HOBO sensor is slower than the Campbell, but coefficient of 1, and then conclusion is that there is a lack of consistent pattern. I do not follow this, please explain and rephrase
*As stated in the text: 5 minute data show that the HOBO sensors are slower, but hourly averages have a correlation of 1. Figure 7 shows the lack of consistent pattern. Added references to figures in the conclusion sentence: "The lack of a consistent pattern in these comparisons (Figure 7) prevented us from adjusting the HOBO temperature data to match the Campbell data. The high correlation and low temperature offsets (Figure 8) among sensors gave us confidence that using HOBO stations to assess temperature patterns across the basin was valid."*

P13L20: add some explanation/conclusion.  I miss in this whole section why you do the comparison between the sensors and eventually the physical interpretation or conclusion from your statements.
*Added explanation to the top of the section and this physical interpretation to this paragraph: "Again, this gave us confidence that we could use our data to assess humidity variations across the basin."*

P14L1-2: did more people had this problem? Is it a random tip that can also occur during dry periods (since this can not be filtered out)? Or is the tip sometimes 'stuck'?
*We don't know of others that have run into this problem. It is not a random tip because when the data show rain at one station, there is almost always also rain at other stations. We never observed the bucket getting stuck in our calibration.*

P14L4: or conclusion is the HOBO has a sensor problem.

*Yes that is the conclusion of the previous paragraph. This paragraph focuses on the Campbell sensor and the conclusion is that if the internal electronics create a double tip, the logger is filtering it out.*

P14L16: is this katabatic flow measured or a assumption it develops?
*We measured wind speeds and directions at the On-Ice station consistent with a katabatic flow in summer, as the text now notes.*

P15 Figure 7: include the colours in the caption
*Added "Each colored line represents a HOBO sensor, blue dots are for the reference Rotronic sensor, and red dots are for a Campbell 107 Temperature Probe."*

P16-P17: Section 3.3 I do not think this section is a great addition to your purpose of the paper and not supported by any in depth discussion, please remove.
*We included this section to help explain differences in glacier mass balance discussed later. For the revisions, I added text to the mass balance section to make the connection more direct. "These mass balance patterns are consistent with the warming temperatures and relatively stable precipitation measured at Talkeetna Airport."*

P18 Figure 11: is this the same transect measured every week at same location? What do you mean with "plotted relative to a reference station"? Does this mean steepness in line is varying in time? Please explain in caption. Mention in caption what upper stations in winter are not operating/measurement problems.
*The transect is not the same in every week. The stations are fixed in place, and whenever the station has enough data to calculate a weekly average, it gets plotted. Changed caption to: "Weekly air temperature profiles show the winter inversions and summer differences between glacier and tundra temperature. For each week, the reference station (Windy Creek Lower, 940 m a.s.l., triangles and black dots) was plotted on the horizontal axis according to the date. The other stations (triangles or circles) were plotted to the left or right of the reference station according to their temperature difference."*

P19 figure 12: Add coloured lines for each of the dots to show whether the gradients change in time/how sensitive they are.
*Given the inconsistent recovery of data from different sites in different years, we chose not to display best fit lines. It doesn't make statistical sense to try to compare these lines from year to year.*

P19 figure 12: Add the resolution of the glacier inventory in the caption.
*Vector outlines don't really have a resolution. The reader can see the detail of the outlines in Figure 1.*

P19 figure 12: Insert in caption how the mass balance in computed (from the "HOBOglacier" station in Table A1?
*Inserted "...measured with the glaciological method"*

P19L1-2: or the measurements are not representative for the whole region.
*Added "... or the best-fit line might be too sensitive to data from lower elevations."*

P19L4-end: the linear interpolation is done with all data such that no division is made between years (1 average value for all glaciers and all times?)? If so this is a very simplified method and I am not convinced in the numbers you present. For example P is highly spatially variable, as you also state in P20L14-15.
*Added "... for each year"*

P20 Section 4.2 Very limited, please expand or consider removing or merging this section.
*It might not take much space, but this section includes some key results. Reworded a few sentences to make it read better.*

P20L17: I do not follow this, you did not do any model simulation of snow accumulation, or this is not mentioned.
*Reworded to emphasize comparison to our stake network.*

P21L2-3 please rephrase. Absolute difference at 2000 and 1000m or did you do some averaging?
*This is an approximate estimate based on binned radar data and point measurements for multiple years and multiple glaciers. We're trying to convey the big picture here, and the reader can refer to the figure for the numerical details.*

P21L4: how do you know surface roughness is responsible? Please add explanation or supporting material for this statement.
*Added: "Over short spatial scales in the ablation zone, surface roughness is responsible for high spatial variability in SWE. The end-of-summer glacier surface is rough due to streams, crevasses, melt ponds, and moraine material. The end-of-winter snow surface tends to be relatively smooth compared to the summer surface, but can also have wind-derived roughness features that contribute to the variability in SWE over small distances."*

P21L6-8 I am not convinced
*What do you think it is then? We tried our best to adequately account for errors in our methods and this lists what we couldn't account for.*

P22L9: again, why the roughness of the ice surface?
*See above*

P22L16: add explanation why data become noisier. Why does ice give more noise signal?
*Added: This was likely caused when the acoustic signal bounced off different elements of the rough ice surface in successive measurements.*

P22L20: you assumed constant density? What are the implications of this assumption?
*Changed "assumed" to "used" since this is based on measurements. New sentence reads: "To calculate ablation from the observed distance change, we used a density of 350 kg m$^{-3}$, based on the average of 5 snow density measurements at the site."*

P24 Table 6: increase fontsize
*Table doesn't fit on the page with a larger font. The copy editors may be able to fudge this somehow.*

Section 5.2: please remove, I do not think is Section is of additional value
*Readers who are familiar with the prior work or the field site will be interested in this information.*

P24L11: "though we did not do a detailed texture analysis", But still you know it matches with the STATSGO soil map? I am not convinced.
*Soil texture is relatively easy to observe in the field and the soils we encountered were largely sandy loam. This matches with the STATSGO map. Sites with very little soil development (Windy Creek Upper, Valdez, Two Plate) show as bedrock on the STATSGO map. We also looked into some of the STATSGO attributes such as organic content and drainage characteristics, which aligned broadly with our field observations. As mentioned, we did not do a more detailed texture analysis (e.g. taking samples back to the lab), so what more do you want to convince you?*
*Changed: "Though we did not do a detailed texture analysis on the soils, the characteristics we observed generally matched up with the State Soil Geographic (STATSGO) soil map (https://datagateway.nrcs.usda.gov/)."*
*To: "Though we did not do a detailed texture analysis on the soils, our observations of soil texture, organic content, and drainage characteristics generally matched up with the State Soil Geographic (STATSGO) soil map (https://datagateway.nrcs.usda.gov/)."*

P25 Section 6.2 Explain the uncertainties and effect by the disturbance of the disturbed soil on the measurements.
*Added: "By pushing the sensors into the undisturbed soil on the side of the pit, we hope to minimize any temperature errors due to disturbance. We did not quantify the effects of disturbance on temperature, though we expect them to be small compared to the large temperature changes observed over the record. Deep sensors were deployed at the bottom of a 2.5*

*cm-wide hole drilled into the ground. Disturbance of the soil around the deep sensors may affect temperature too, but we tried to minimize that by using the smallest drill we could."*

P27L11-13: why is this relevant, please remove.
*Unclear what the reviewer thinks is not relevant - this whole paragraph is describing the data in figure 18.*

P27L25: How do you relate this to the dam? No runoff analysis is done.
*Deleted reference to dam here.*

P27L27: this is only a minor section and for me not strong part of your paper and now you present the climate change numbers as one of your main conclusions.
*I think it is totally fair to present our data as a baseline for future measurements. The point of concluding statements is to highlight how this paper and the data within it are relevant to the broadest audience possible.*

P28 Figure 18: air temperature is gray colour?
*Yes, as indicated by the legend.*

P28L6: not new conclusion, snow amounts are generally higher at high elevations
*True, but it has not been shown for this study area and time period, so it is still a valid conclusion.*

P28L7: you did not measure the soil and your conclusion is that these match with the mapped soil descriptions. Please remove this statement out of your conclusions and preferably also out of the text
*Detail added above to support these conclusions.*

Conclusion in general: please do not focus on climate change and dam implications, but give conclusions about the data you found in the field. What did you find and why is it special?
*We do give conclusions about our data and we highlight how the data are relevant to the broadest audience possible (people who are interested in climate change and runoff).*

---

## Author Comment (AC2) · 9 Oct 2019

**RC2:**

General Comments:

This study presents, validates and interprets a comprehensive and impressive data set, which covers a range of parameters in the variable environments of the Susitna Basin, Alaska. The data set includes meteorological, glaciological and soil parameters. The data set is unique, as many of the measurements were done in complex terrain where measurements generally are sparse. It is effortful and requires extensive planning to acquire meaningful data in this terrain. Problems in the data are addressed and generally, implications that arise with these problems are described in detail. Overall, the manuscript is well structured and provides a good overview of the data. The data set itself could be extremely valuable for model validation or comparison with future field studies.

***Thanks!***

However, the manuscript is not always coherent and suffers from redundant information (e.g. section 3.3, section 5.2, figure 9),

***None of these sections or figure are redundant. Perhaps the reviewer meant to say relevant? That critique was addressed in response to reviewer 1 and can be summarized with: "Readers who are familiar with the prior work or the field site will be interested in this information."***

which distracts the reader from following the key points and weakens the focus of the paper (see specific comments).

***Addressed below***

The introduction does not adequately motivate the manuscript, as it does not really make clear why the data set is important and what the purpose of the data set is (see specific comments).

***Addressed below***

In addition, some of the presented data appear isolated and need to be put into context better (e.g. section 4.4., how do continuous mass balance measurements compare to stake measurements nearby?) ***See P22L10: "The net depth of snow lost at the sonic distance sensor site was 1.97 m (17 April - 30 June), which is comparable to the 2.15 m snow depth measured in a snow pit about 6 m away on 14 April 2013." And see figure 5 which shows the ablation stake and continuous measurements together. Added reference to figure 5 in section 4.4.***

Therefore, I suggest a number of minor revisions to focus the main messages of the manuscript and to emphasize the uniqueness of the data set.

Specific Comments:

Introduction:
Mentioning climate change in the beginning of the introduction is not convincing, since you only acquire three years of data. Either remove the link to climate change, or emphasize that the data set is meant to be used for comparison with future studies (as you do in the conclusions).

***We also compare our data to data from 30 years ago and see evidence of climate changes, so we assert that climate change is in fact relevant to the paper.***

P2, l9-10: using changes in river flow on dam operations as a motivation here seems misplaced, since you mention in the beginning that the dam was not built. In addition, river flow is not covered in this study. Please rephrase or remove. You could motivate each of the data types (meteorological/climatological, glaciological, snow, soil) individually, as you do later in the manuscript. E.g., p4,l23-26 motivates meteorological/climatological measurements, p25,4-5 motivates soil measurements and should be placed in the introduction

***This was the main motivation for the project (and the reason the project was funded) so we believe it to be relevant context. The wording makes it clear that the paper focuses on the data, this section of the intro is providing the context for why the data was collected. We added a sentence about data scarcity to bolster the motivations.***

- p1, l6: since only the years 1981-83 were investigated, please do not write 1980s here but refer specifically to the years 1981-83
*DONE*

- p2, l5: state precise number instead of "more than 120 glaciers"
*Changed to "more than 100", see notes to reviewer 1*

- p3, l5: please add reference here
*Added: Scherler, Dirk, Bodo Bookhagen, and Manfred R. Strecker. "Spatially variable response of Himalayan glaciers to climate change affected by debris cover." Nature geoscience 4.3 (2011): 156.*

- p3, l10-p4,l3: detailed description of surge history seems unnecessary here, please shorten
*We focus on mass balance, which can be affected by surging (and vice versa). The latest surge of West Fork occurred shortly after the 1981-83 mass balance measurements to which we are comparing our data. While the effect of a surge on the mass balance may not be quantifiable given the data that is available, the surge still needs to be mentioned.*

- p6, l4: what does the number 3 in brackets mean?
*Added "Figure" to the 3.*

- p6, l26: first time abbreviation "DGGS" is mentioned, please provide full name
*Spelled out: Alaska Division of Geological & Geophysical Surveys*

- p7, figure 4: please provide figure in higher resolution
*Done*

- p8, l13: delete sentence "Data are not available from 18 January 2014 to 22 April 2014 when the station was buried in snow." since this was mentioned before.
*Done*

- p9, table 2 caption: delete "18 January 2014 to 22 April 2014." from caption, since it was mentioned before
*Because readers may take values from the table without closely reading the text, we believe this mention should remain.*

- p11, figure 5: please provide higher resolution figure
*Done*

- p12, figure 6: figure is not mentioned in text, please add reference to figure in text. Also, please provide figure in higher resolution
*Added reference and higher resolution figure*

- p13, l8: reference to figure 4: is this right or do you mean figure 7?
*Good catch, thank you*

- p13, l11: rather say "very close to 1.0"
*Point taken that it is not exactly 1.0000000. But with two significant digits, I think it is appropriate to report it as 1.0 rather than "close to 1.0" implying it could have been 0.99.*

- p13, l12/13: "The lack of a consistent pattern in these comparisons prevented us from adjusting the HOBO temperature data to match the Campbell data." This is confusing since you mention an average offset the sentence before. Can you clarify this?

*Added references to figures to clarify: "The lack of a consistent pattern in these comparisons (Figure 7) prevented us from adjusting the HOBO temperature data to match the Campbell data. The high correlation and low temperature offsets (Figure 8) among sensors gave us confidence that using HOBO stations to assess temperature patterns across the basin was valid."*

- p.14, l1/2: What is your confidence that no double tips are missed or that normal tips are identified as double tips?
*For a complete discussion, please see Wolken et at 2015. Reference added here.*

- p15, figure 7: how do you explain very high RH offset of some HOBO-sensor at higher RH, especially in 2013?
*Simply different sensor performance, as mentioned in the text. It is notoriously difficult to accurately measure humidity at cold temperatures (e.g. -10 C).*

- p14, l11: "Precipitation amounts did not correlate significantly with elevation, slope, aspect, or location." How do you define "location"? What drives the variations in precipitation amounts?
*Location can be defined with lat/long coordinates, or by logical grouping in mountain ranges. We chose not to speculate on what drives variation, but we can rule out the factors mentioned.*

- p14, l15-17: katabatic wind flow: Can you back this with references or add a more thorough analysis based on your data, e.g. wind direction analyses? Or is this just an assumption you make?
*Added a paragraph on wind direction to the "Meteorological data" section.*

- p15, figure 7: colour codes are missing
*Added to caption: Each colored line represents a HOBO sensor, blue dots are for the reference Rotronic sensor, and red dots are for a Campbell 107 Temperature Probe.*

- p16, figure 9: what is the purpose of figure 9? There is no in-depth analysis provided in the text and patterns are trivial (yearly temperature cycle, lower temperatures with higher elevation). In addition, it figure again suffers from relatively low resolution. Please either remove figure or provide more detailed analysis
*We believe the patterns are not trivial, but chose to spend more space in the paper describing mass balance data. Future papers may delve deeper into the spatial and temporal variations introduced here. We also feel it is best to present this "raw" data before looking at it in terms of lapse rates (figures 10 and 11). Fixed resolution issue.*

- p16, section 3.3: this section does not provide a thorough analysis and is not useful for the manuscript since it is not based on your data. Either please remove or transform; rather than a trend analysis, the section could provide an assessment whether the years 2013-2014 were exceptional (in terms of temperature) or normal.
*We later reference this section when discussing mass balance differences between the 1980s and recent periods.*

- p18, figure 11: what purpose do the different colours serve? If none, please use black or dark grey
*The colors help distinguish which data goes with each week. A grayscale version of this figure is illegible.*

- p19, l12: Did you think about adding the line fit to the figure? This would allow the reader to clearly identify the equilibrium line altitude you derived
*Given the inconsistent recovery of data from different sites in different years, we chose not to display best fit lines. It doesn't make statistical sense to try to compare these lines from year to year.*

- p19, l1/2: Since you used only point measurements, it is also possible that the stake measurements do not fully represent the area that was covered by satellite. In addition, you mentioned earlier that most of the stakes were placed on the centreline of the glacier, which is typically higher than the margins (which are included in the satellite estimation?), potentially leading to higher estimation of the equilibrium line altitude
*Reworded*

- p19, l7/8: "Glacier-wide mass balance estimates were then calculated by summing the distributed mass balance over the whole glacier." Did you use hypsometry of each individual glacier? Or did you use hypsometry of the entire glacier area for the calculation of the individual glacier mass balances? If so, the numbers you get probably have very high uncertainties. Please clarify.
*Changed "based on the glacier hypsometry " to "based on the individual glacier's hypsometry"*

- p20, l2: "East Fork Glacier had a similar mass balance as Maclaren Glacier." Why do you stress this here? Seems misplaced, please delete
 *Done*

- p20, l17: "To robustly validate model simulations of snow accumulation…"; please move to introduction, since this provides a motivation for your measurements and you are presenting results in this section
*Removed reference to modeling as suggested by reviewer 1.*

- p21, l2/3: "…at 2000 m measuring 2-3 times higher than at 1000 m." looks more like 2-4 times higher (see year 2014)
*True, changed to 2-4.*

- p21, l4: "A notable south-north decrease in total SWE and accumulation gradient indicates a strong orographic influence." Please remove: sentence is redundant since you mention elevation dependence the sentence before. In addition, this is not always necessarily a north-south gradient
*Done*

- p22, figure 14 caption: "In early August 2013, the sensor's mounting pole began to tip over and give bad readings. On 1-2 September 2014, 18 cm of snow accumulation was recorded, consistent with observations during a site visit. The sensor pinged off falling snow, so some points in that window are labeled as bad data." please remove or move to text
*I think it is valid to leave this short description of the bad data in the figure caption, so the viewer doesn't have to search the text for the description of the bad data.*

- p22, l16: "data became noisier as the surface transitioned from snow to ice." can this be seen in figure 14?
*Yes, when zoomed in.*
- p22, l21/22: "This leads to an average melt rate of 0.016 m w.e. d−1 for the summer of 2013 and 0.012 m w.e. d−1 for 2014." This information seems a bit isolated from the previous, interesting mass balance investigations. Can you provide a comparison here? How does this compare to nearby ablation stakes summer mass balances?
*The previous paragraph mentions this comparison, and it is plotted in figure 5 (added reference here). "The net depth of snow lost at the sonic distance sensor site was 1.97 m (17 April - 30 June), which is comparable to the 2.15 m snow depth measured in a snow pit about 6 m away on 14 April 2013."*

- p23, l7: "Snow water equivalent (SWE)": abbreviation has been initialized before
*Deleted "SWE"*

- p23, l14: "…generally showed a strong elevation dependence." dependence of what type? Maybe just write "increased with elevation"
 *Changed*

- p23, l16/17: "At the Lower Windy Cr. site, about 40 mm of SWE (25%) was lost due to melt of the end-of-winter snowpack between 9 April and 22 April 2014 (Table 6)" Why is this stressed here? Please remove
*Added context: "Lower Windy Creek was the only site where we collected SWE data more than once in a year. About 40 mm of SWE (25\%) was lost due to melt of the end-of-winter snowpack between 9 April and 22 April 2014 (Table \ref{tab:Snow})."*

- p24, section 5.2: This section does not add any value to the paper but is very distracting; please remove
*Readers who are familiar with the prior work or the field site will be interested in this information.*

- p25, l1: "characteristics we observed"; please specify these characteristics so the agreement between soil pits and STATSGO becomes clearer
*Done: "our observations of soil texture, organic content, and drainage characteristics"*

- p25, l4/5: "Understanding the distribution of permafrost and seasonally-frozen ground across the basin is important for modeling of water moving across the landscape"; again, this provides a motivation for your measurements and you are presenting results in this section, so please move to introduction

- p27, l28/29: "Summer air temperatures in 2012-2014 were 1.1∘C warmer than 1981-1983. Annual temperatures were 0.5°C warmer in the recent period." Why do you add this here? This is not based on your own measurements and thus not a significant outcome. Please consider removing.
*It is significant because it helps explain the glacial mass balance changes we observe. Since we do not do temperature index modeling in this paper, we do not directly attribute the mass balance change to the temperature change, though we assume (hope!) the reader will see the connection.*

- p28, l2: since only the years 2012-14 vs. 1981-83 were investigated, please do not generally say 2010s vs. 1980s since you have no information on the other years
*Done*

Technical Corrections:
- p8, l12: move "On-Ice" to the end of this sentence
*Done*
- p6, l33: please remove "instead of every minute"
*Done*
- p16, l4: "most distinct" instead of "least complicated"
*Done*
- p18, l2: "refers to the period October…" instead of "refers to October…"
*Done*
- p20, l11/12: "Therefore, lower annual balance in the latter period (-1.72±0.87 m w.e.) compared to the former period (0.04±0.25 m w.e.) were driven by the more negative summer balances."
should be "balances… were driven…" or "balance… was driven"
*Done*
- p26, figure 17 caption: "soil" instead of "soils"; remove "," after data
*Done*

---

## Editor Decision (ED1)

Dear Andrew et al,

I have read your manuscript and especially focused on your answers to the referee comments. All in all, I appreciate the detailed description of the instruments, their calibration and the resulting data. However, I do agree with the key comments of the reviewers (below) and would like to ask you to address them a little more than in the revised version.

(1)

Both reviewers suggested that parts of the introduction (i.e. the dam project and the relevance of the data for climate change and river runoff) should be modified and the introduction more focused on the relevance of the data for the validation of modeling input data. You answered that the more global context is relevant for readers beyond the community. This is correct, however, the introduction shall not raise expectations that are not fulfilled in the later parts of the manuscript.

I do understand your point with the dam that was the funding reason for this project and think it is ok to keep it as you suggested.

However, reading your manuscript, "we performed extensive field measurements in the same area. Our work combined field measurements with glacier runoff modeling to make projections of the effect of climate  change induced future glacier mass changes on the inflow to the proposed dam; this paper focuses on the measurements" gives me the impression (and I am speaking as out of your community, I am a structural geologist by training and data curator for the last 5 years) that you are describing the data here (this paper focuses on the measurements) and that the glacier runoff models including the projections of the effect of climate change induced changes (etc) are part of another study?

If my interpretation is correct, I think that it would make everything much clearer if you stated this clearly in the introduction (even if the modeling paper is not yet published). What I mean is to slightly expand the last half sentence from "; this paper focuses on the measurements" to ". This paper focuses on the measurements while the resulting glacier runoff models etc... will be described in future studies..." (or similar).

(2)

My second question related to the old data from the 1980s... are these available?

I have glimpsed through the cited papers with the following result:

- Clarke 1991 (please add https://doi.org/10.3189/S0022143000042842 to the reference in the manuscript): no data present, except for plots and summary data like annual sliding velocity, surface ice flux etc...
- Clarke et al, 1985: data tables ... are these the data you are referring to?
- what about the two R&M consultants reports. I don't find online versions of them, only citations and the index of reports of the Susitna hydroelectric project. Are the cited reports online available? The titles are mentioning data....

It seems as at least some data are available as printed pdf (sometimes unfortunately covered by a paper note as in one table in Clarke et al, 1985) - do you possibly have some digital versions of the data (or could you imagine making them, i.e. extracting the data from the report or pdf) and could make them available? This would definitely involve some work (and the agreement of the authors of the data), but having some reference data from the 1980s in digital form would be so important, don't you think so? Of course, I am not requesting this, I am just asking if it would be possible? Especially the old reports (that are not online) could be copied with OHS and the data extracted.... I have no idea how many tables are in the reports and would focus on measured data whenever possible....

---

## Author Response (AR2)

I (Andrew Bliss) received comments in an email from Kirsten Elger, the editor for this article. Her attachment with the comments did not come through in the Copernicus system so I've pasted them here. Our responses are in blue text below.

(1)

I have read your manuscript and especially focused on your answers to the referee comments. All in all, I appreciate the detailed description of the instruments, their calibration and the resulting data. However, I do agree with the key comments of the reviewers (below) and would like to ask you to address them a little more than in the revised version.

Both reviewers suggested that parts of the introduction (i.e. the dam project and the relevance of the data for climate change and river runoff) should be modified and the introduction more focused on the relevance of the data for the validation of modeling input data. You answered that the more global context is relevant for readers beyond the community. This is correct, however, the introduction shall not raise expectations that are not fulfilled in the later parts of the manuscript.

The revised manuscript does not raise expectations that are unfulfilled. I deleted the mention of the dam in the abstract. Prior to resubmission we already reduced the focus on climate and the dam. The introduction is supposed to provide a little context and some broader relevance - that is all that is left.

I do understand your point with the dam that was the funding reason for this project and think it is ok to keep it as you suggested.

Howver, reading your manuscript,

"we performed extensive field measurements in the same area. Our work combined field measurements with glacier runoff modeling to make projections of the effect of climate change induced future glacier mass changes on the inflow to the proposed dam; this paper focuses on the measurements"

gives me the impression (and I am speaking as out of your community, I am a structural geologist by training and data curator for the last 5 years) that you are describing the data here (this paper focuses on the measurements) and that the glacier runoff models including the projections of the effect of climate change induced changes (etc) are part of an other study?

Yes, that is correct.

If my interpretation is correct, I think that it would make everything much clearer if you stated this clearly in the introduction (even if the modeling paper is not yet published). What I mean is to slightly expand the last half sentence from

"; this paper focuses on the measurements"

to

". This paper focuses on the measurements while the resulting glacier runoff models etc... will be described in future studies..." (or similar).

This small change would make much clearer, don't you think so?

Changed, as suggested, to: "This paper focuses on the measurements, while the modeling results have been described in Wolken et al. (2015)."

(2)

My second question related to the old data from the 1980s... are these available?

I have glimpsed through the cited papers with the following result:

- Clarke 1991 (please add https://doi.org/10.3189/S0022143000042842 to the reference in the manuscript): no data present, except for plots and summary data like annual sliding velocity, surface ice flux etc...

Added DOI

- Clarke et al, 1985: data tables ... are these the data you are referring to?

Not really, primarily R&M - see below

- what about the two R&M consultants reports. I don't find online versions of them, only citations and the index of reports of the Susitna hydroelectric project. Are the cited reports online available? The titles are metioning data....

Yes - but they are well-hidden online. I added the links to the reference list. https://www.arlis.org/susitnadocfinder/Record/375325 and https://www.arlis.org/susitnadocfinder/Record/375326

It seems as at least some data are available as printed pdf (sometimes unfortunately covered by a paper note as in one table in Clarke et al, 1985)... do you possibly have some digital versions of the data (or could you imagine making them, i.e. extracting the data from the report or pdf) and could make them available? This would definitley involve some work (and the agreement of the authors of the data), but having some reference data from the 1980s in digital form would be so important, don't you think so? Of course, I am not requesting this, I am just asking if it would be possible? Especially the old reports (that are not online) could be copied with OHS and the data extracted.... I have no idea how many tables are in the reports and would focus on measured data whenever possible....

this is really just a conning idea, but nevertheless. I wanted to address it...

We have converted the data from the pdf to machine-readable forms. The 1980's mass balance data is already included with the modern data in the data repository. However, we would prefer not to republish additional data (e.g. snow survey data, weather data) this since it is not our own data.

**Please let me know what you think about it?**

Thank you for your careful attention to our paper. I hope that our comments and edits have sufficiently addressed your concerns.